# Iron-catalyzed aliphatic C−H functionalization to construct carbon−carbon bonds

Lulu Zhou[1,2,3], Hengrui Cai[1,2,3], Dong Xie[1,2,3], Kangkang Sun [1,2,3], Shanmei Zhu[1,2], Mengying Guo[1,2] & Wei Han [1,2] ✉

Although cytochrome P450 enzymes are powerful catalysts for hydrogen-atom abstraction from alkanes by iron-oxo species, the process typically leads to oxygenated products due to ultrafast oxygen rebound. Developing synthetic catalysts that mimic this activity while avoiding oxygenation remains challenging, especially for intermolecular carbon−carbon bond formation. Here, we report an iron/bioinspired ligand catalyst that uses hydrogen peroxide to enable undirected methylene C−H functionalization with 1,4-quinones and azines, allowing direct formation of medicinally relevant C−C bonds while suppressing oxygen rebound. The reactions proceed efficiently with two equivalents of diverse alkanes, and the site selectivities, which differ from those observed in traditional methods, can be predicted based on steric, electronic, and stereoelectronic effects, even in complex molecules. This catalyst overcomes the intrinsic limitation of P450s, which favor oxygen incorporation over free radical formation, offering a promising strategy for selective alkylation of quinones and heterocycles using feedstock alkanes.

Selective functionalization of aliphatic C−H bonds is changing synthetic chemistry because it can install a synthetically useful group into inherently unreactive yet inexpensive feedstock chemicals by substitution of the most ubiquitous moiety in an organic molecule, representing an ideal approach to molecular upgrading[1,2]. Nevertheless, the undirected functionalization of strong, neutral C($sp^3$)−H bonds is arguably the greatest challenge[3–5]. Recently, impressive developments in hydrogen atom transfer (HAT) strategy have enabled the elaboration of aliphatic C−H bonds to be achieved using stoichiometric hydrogen-atom transfer (HAT) agents[6,7]. Furthermore, an appealing photocatalyzed HAT have been successfully tested[8–15], but the typical requirement for the substrate to be the solvent or in large excess (often >5 equivalents). Alternatively, the methods involve formation of metal−carbon bonds for the functionalizations of alkanes. For instance, Intermolecular alkylation of C−H bonds can be achieved by utilizing rhodium catalysis[16,17]. The transformation however, suffers from restricted substrate scope due to the use of donor-acceptor diazo

reagents for alkylation and the limited accessibility of the starting materials. As an elegant example, iridium catalysis enables undirected borylation[18] with alkanes as limiting reagent, despite the high cost of using a precious metal, which limits its large-scale applications. Notwithstanding these advances, the use of earth-abundant metals in catalysis for the undirected functionalization of strong alkyl C−H bonds with substrate as the limiting reagent or in minimal excess remains a grand challenge.

In nature, enzymes such as heme-thiolate monooxygenase cytochromes P450 are ideal candidates to address unmet reactivity and selectivity challenges in undirected inert aliphatic C−H functionalization[19]. The catalysis capability of the biocatalyst is realized by a reactive high-valent iron-oxo intermediate. The high-valent iron-oxo species abstracts a hydrogen atom from an alkane substrate followed by oxygen rebound at a typical rate constant between $10^{10}$ and $10^{12}\,s^{-1}$ (Fig. 1A). Due to the ultrafast oxygen transfer rates, reactions involving high-valent iron-oxo intermediates would inevitably result in

[1]State Key Laboratory of Microbial Technology, Jiangsu Collaborative Innovation Center of Biomedical Functional Materials, Jiangsu Key Laboratory of Biofunctional Materials, Jiangsu Key Laboratory of New Power Batteries, Nanjing Normal University, Wenyuan Road No.1, 210023 Nanjing, China. [2]School of Chemistry and Materials Science, Nanjing Normal University, Wenyuan Road No.1, 210023 Nanjing, China. [3]These authors contributed equally: Lulu Zhou, Hengrui Cai, Dong Xie, Kangkang Sun. ✉e-mail: whhanwei@outlook.com

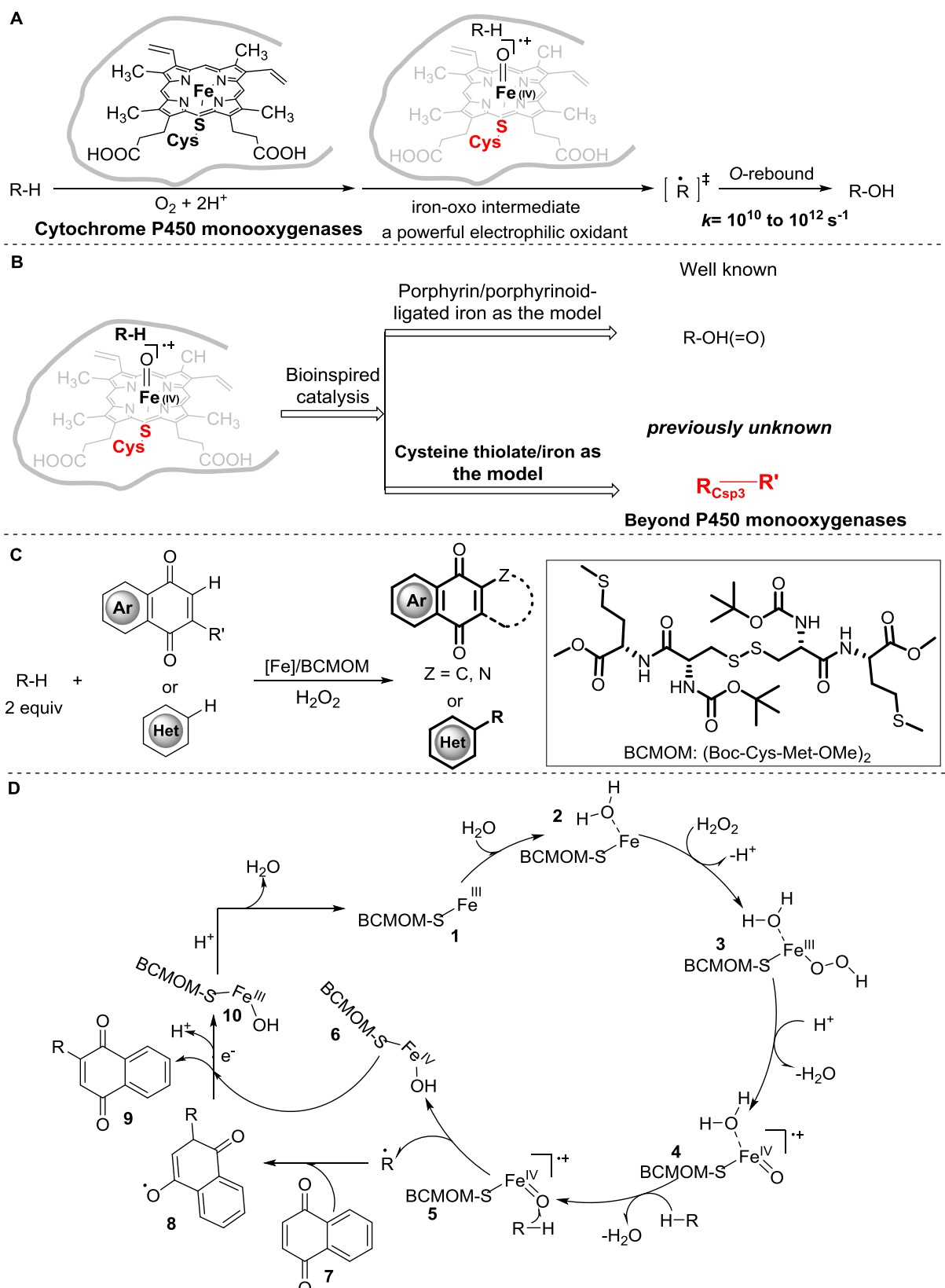

**Fig. 1 | Bioinspired alkylation process using alkanes as alkylating agents.**
**A** Cytochrome P450 monooxygenases abstract H-atom from alkane followed by
O-rebound to give the corresponding alcohol (*k*, rate constants for oxygen

rebound); **B** cytochrome P450 monooxygenases inspired catalysis; **C** bioinspired
iron catalyzed direct installation of alkyl groups using alkanes enabled by a disulfide
ligand BCMOM; **D** proposed mechanism.

the generation of oxygenated products[20,21]. Indeed, the cytochromes P450 are renowned as highly selective biocatalysts for remarkable aliphatic C−H hydroxylations[22], and biomimetic heme/non-heme iron catalysis involving high-valent iron-oxo species is commonly limited to oxygenation reactions (Fig. 1B)[23–26]. Moreover, only rare reports disclosed that the incorporation of other functional groups (Cl, Br, N₃) into C−H bonds required stoichiometric amounts of non-heme iron-oxo complexes[27,28]. Although an example of catalytic non-heme iron complexes to mimic the function of iron halogenases can realized chlorination of $C_{sp3}$−H chlorination, the strategy is limited to the substrates bearing activated $C_{sp3}$−H bonds and provides low yields of the corresponding chlorinated products along with substantial oxygenation side reactions[29]. Implementing high-valent iron-oxo as the catalytic species for efficient alkane functionalization while avoiding oxygenation remains a significant challenge, especially in intermolecular carbon−carbon bond formation reactions (Fig. 1B). Unlike heteroatoms, carbon is difficult to coordinate with a transition-metal catalyst and is prone to oxygenation under oxidation conditions, even if a carbon-metal species is formed. For instance, heteroatom-rebound catalysis (HRC) strategy involving high-valent porphyrin-Mn-oxo intermediates are restricted to $C_{sp3}$−H halogenation and nitrogenation[25,30].

Quinones and nitrogen-containing heterocycles are ubiquitous chemical scaffolds with significant biological and pharmaceutical activity. Given their influence on drug metabolism and pharmacokinetic profiles[31,32], there is growing demand for the direct installation of alkyl groups into quinones and heteroarenes, especially for the processes using normal stoichiometry of alkanes as alkylating agents. Unfortunately, most current methods available for these reactions require the substrate to be the solvent or in large excess (>5 equivalents) and are ill-suited to the intermolecular functionalization of complex molecules. We sought to introduce cytochrome P450 enzymes-guided iron catalysis for alkylation of 1,4-quinones and azines including complex substrates with substrate in two equivalents. For our design, we drew inspiration from prior research that electron-donating property of a thiolate ligand may serve not only to enhance the H atom abstraction ability of the iron-oxo but also to suppress oxo-transfer reactivity[33,34].

Here, we report that a thiolate-ligated iron−oxo favors hydrogen-atom abstraction over oxygen-atom transfer to obtain an escaped alkyl radical followed by addition to 1,4-quinones and azines (Fig. 1C, D), a process for which catalysts have long been desired.

## Results and Discussion
### Reaction development
The cysteine thiolate axial ligand of the heme iron center in natural P450 enzymes is generally accepted to govern the oxidative reactivity of these cytochromes and promote C−H bond cleavage[35]. However, this property used for bioinspired catalysis remains quite rare[36]. In addition, a thiolate ligand can enhance the H atom abstraction ability of the iron−oxo while suppressing oxygen-rebound pathway[33,34], but the use of the character to design catalysts for organic synthesis has been overlooked probably due to sulfur atom generally believed to readily poison transition metal catalysts. Inspired by these findings, we developed thiolate ligands based on the cysteine to obtain P450-like catalysis and discovered BCMOM as the optimal ligand that enhances iron-oxo abstracting hydrogen atom ability but suppresses its oxo-transfer reactivity with H₂O₂ as oxidant in CH₃CN/H₂O (Fig. 2 and the detailed reaction optimization in Table S1), thus realizing highly selective alkylation of 1,4-quinones and azines with substrate in two equivalents (Fig. 2C). The BCMOM/Fe catalyst is highly responsive to electronic effects, highlighting the regioselectivity of it for more hydridic C−H bonds imparted by the electrophilic nature of its iron-oxo[23,24]. When the BCMOM/Fe catalyst was used for oxidation of arene C-H (a specific acetanilide) with H₂O₂ in CH₃CN/H₂O[18], O¹⁸ incorporation into the desired product was observed. The presence of the

thiolate ligand significantly stabilizes the iron center and lowers the redox potential of the iron catalyst, facilitating the formation of high-valent iron-oxo species[37,38]. More importantly, the formation of a high-valent iron-oxo [(acac)₂(BCMOM)₂·⁺Feᴵⱽ(O)] species was directly confirmed by high-resolution mass spectrometry (Supplementary Fig. 14), further supporting that the active oxidant is a high-valent iron-oxo species [often iron(IV)-oxo][39]. Notably, the highly electrophilic oxidant generated with BCMOM/Fe and H₂O₂ allows for selective abstraction of the most challenging secondary C−H bonds with synthetically useful yields in the absence of directing or activating groups. An intermolecular kinetic isotope effect (KIE) determined by using a mixture of cyclohexane and [D₁₂]-cyclohexane was 2.05 (Supplementary Fig. 8), whereas a bioinspired heme iron catalyst resulted in a relatively high kinetic isotopic effect (KIE > 3)[40], suggesting that our bioinspired BCMOM/Fe catalyst could more effectively facilitate cleavage of inert C−H bonds than classical heme iron, consistent with previous study of the role of thiolate ligation in cytochrome P450[35]. Hydrogen atom transfer (HAT) step provides alkyl radical intermediates that can be trapped by radical scavenger BrCCl₃, indicating a dominant radical escape over rebound[41] for further functionalization as outlined in Fig. 1D[42]. Site selectivity advantages for secondary C−H bonds may stem from easier steric accessibility compared to tertiary C−H bonds and more electron-rich property compared to primary C−H bonds. Additionally, in ring systems stereoelectronic effects may contribute to achieve high site selectivity in secondary C−H bond functionalization. The site of functionalization with the BCMOM/Fe catalyst can be predicted in complex molecules on the basis of steric, electronic, and stereoelectronic properties of the C−H bonds of simple substrates.

### Alkylation of naphthoquinones with alkanes
We next explored the substrate scope of this method for various simple and complex compounds containing inert and activated C−H bonds for coupling with naphthoquinones using the electrophilic oxidant in-situ generated from BCMOM/Fe and H₂O₂ (Fig. 3). Electron-withdrawing substituents deactivate proximal sites via inductive effects, thus favoring selective functionalization at the most remote and least electronically deactivated site. For instance, 2-hexanone afforded alkylated product **11** in 83% yield ($\gamma$:$\beta$ = 2.3:1), while 5-methyl-2-hexanone resulted in desired product **13** in a lower yield and selectivity (70%, $\gamma$:$\beta$ = 1:1) owing to increased steric hindrance. The one carbon shortened analog of 2-hexanone was functionalized exclusively at the $\beta$ position (**12**, 60% yield). Notably, these substrates contain much weaker $\alpha$-C−H bonds, which are commonly the most reactive in C−H transformations, but these bonds are unreactive in our case due to strong electronic deactivation of the sites $\alpha$ to the carbonyl. Further supporting this trend, 2-heptanone and 2-octanone were also investigated, and both gave mixtures of regioisomers with predominant functionalization at remote $\gamma$ and $\delta$ positions, consistent with the higher reactivity of remote sites less affected by the carbonyl group (Supplementary Fig. 19, **154, 155**). Meanwhile, as the carbon chain length increases, the number of regioisomers also increases, resulting in the formation of complex, intractable isomeric mixtures[25]. Likewise, four- and five-membered ketones were also competent with single $\beta$-regioisomers (**14** and **15**, 32 and 70% yield, respectively); a seven-membered ketone and a polycyclic ketone afforded the major products from functionalization of the most electron-rich methylene sites farthest from the EWG (**18** and **20**). Especially noteworthy, structurally similar substrate containing both traditionally reactive $\alpha$-ketone C−H bonds and secondary benzylic C−H bonds, was preferentially functionalized $\gamma$ positions despite its higher C−H bond dissociation energy (**19**, 65%). In contrast to **18−20**, six-membered ketones show no preference for the more electronically preferred $\gamma$ methylene site over the more proximal $\beta$ site because unfavorable 1,3-diaxial strain at C1 and C3, can be alleviated upon C3 C−H functionalization[43]. In addition, an eight-membered ketone was examined and found to undergo

**Fig. 2 | Fe-catalyzed aliphatic C-H functionalization: Influence of thiolate ligands.** Reaction conditions: 1,4-naphthoquinone (0.25 mmol), 2-hexanone (2.0 equiv.), Fe(acac)$_2$ (5 mol%), ligand (10 mol%), H$_2$O$_2$ (3.0 equiv.), CH$_3$CN:H$_2$O (2 mL:2 mL), 1 h at 80 °C under N$_2$. Isolated yields are given.

functionalization at multiple positions, with a preference for the more electron-rich and sterically accessible $\gamma$-methylene sites (Supplementary Fig. 19, **156**). Moreover, the $\beta$-site selectivity can be significantly improved when using more sterically hindered naphthoquinones as the coupling partners (**46-48**). Collectively, these results illustrate the BCMOM/Fe catalyst's ability to discriminate between hydrogen atoms three and six bonds away from an electron-withdrawing group.

A range of substituted molecules, including acid, ester, bromide, primary alcohol, and nitrile substituents, proved to be good substrates for this chemistry. Functionalization of valeric acid provided **21** (75% yield, 67% selectivity). Similarly, reactions of *n*-hexanoic acid, *n*-heptanoic acid, and *n*-octanoic acid proceeded efficiently to give mixtures of regioisomers with predominant functionalization at more remote methylene positions, consistent with the trend observed for linear aliphatic acids, albeit with more regioisomers (Supplementary Fig. 19, **157–159**). Cyclopropyl radicals, though relatively stable, are known to undergo ring-opening to form allyl radicals under certain oxidative or photochemical conditions. However, in the transformation of 3-cyclopropylpropanoic acid under the BCMOM/Fe catalytic system, the cyclopropyl group was fully retained without any detectable ring-opened product, delivering the desired cyclopropyl-containing product **22** (50% yield). The method successfully preserved sensitive drug-like motifs including cyclopropane structures[44], demonstrating its synthetic utility despite the poor result observed with 4-cyclopropylbutanoic acid. Methyl butyrate and butyronitrile both bearing a three carbons-alkyl chain gave only one observed methylene functionalized products at the most electron-rich and sterically accessible $\beta$ sites distal to the electron-withdrawing moieties (**23** and **32**, 60 and 50% yield, respectively). When the alkyl chains were extended by one carbon in both methyl valerate and valeronitrile substrates, significant improvements in yields were observed (**24** and

**31**, 70 and 70% yield, respectively), albeit in diminished site-selectivity (3:2 to 2:1 $\gamma$:$\beta$). The dual electron-withdrawing group substitution pattern substantially boosted the remote regiochemical control, providing access to target adducts (**25**, **28**, **30**) in 70–75% yields with high selectivity. In addition, selectivity for the $\gamma$-position was improved from 3:1 to 4:1 in switching from the benzoate to 4-nitrobenzoate group (**26-27**). Unprotected isohexanol is functionalized in good yield, but with poor regioselectivities (**45**, 70% yield, 1:0.7 $\delta$:$\gamma$ ratio) due to competing electronic preference for the remote $\delta$ site reaction and steric preference for the remote $\gamma$ site reaction. Cyclohexane, a hydrocarbon with no electronically biased substituents, is a viable substrate as well (**42**, 60% yield).

Considering the biomedical significance of heteroarene and amine functionalities, we evaluated the BCMOM/Fe system with these challenging substrates. Such compounds typically pose difficulties in oxidative C–H functionalization due to (i) competitive coordination of pyridine/amine groups to the catalyst, and (ii) susceptibility to deleterious side reactions including N-oxidation and C–N bond cleavage. We investigated BCMOM/Fe catalysis with protonation of the basic nitrogen atom strategy for rendering remote methylene C–H functionalization of nitrogen heterocycles and amines (Fig. 1C, D)[45,46]. Pyridine is the second most common nitrogen heterocycle, appearing in U.S. FDA approved drugs[44] and its derivatives were tolerated and afforded remote methylene functionalized products in good yields (**34–36**) even in the presence of more reactive benzylic C–H bonds. Using *N,N*-dimethylcyclohexylamine or *N,N*-dimethylbutylamine, substrates that have traditionally reactive $\alpha$-amino C–H bonds, BCMOM/Fe delivered the corresponding remote functionalized products **37–38** in 60-65% yields. Piperidines substituted at N and C4 are the most prevalent nitrogen heterocycles in drugs[47]. 2,2,6,6-Tetramethylpiperidine and *N*-phenylpiperidine both afforded the C4-functionalized products as

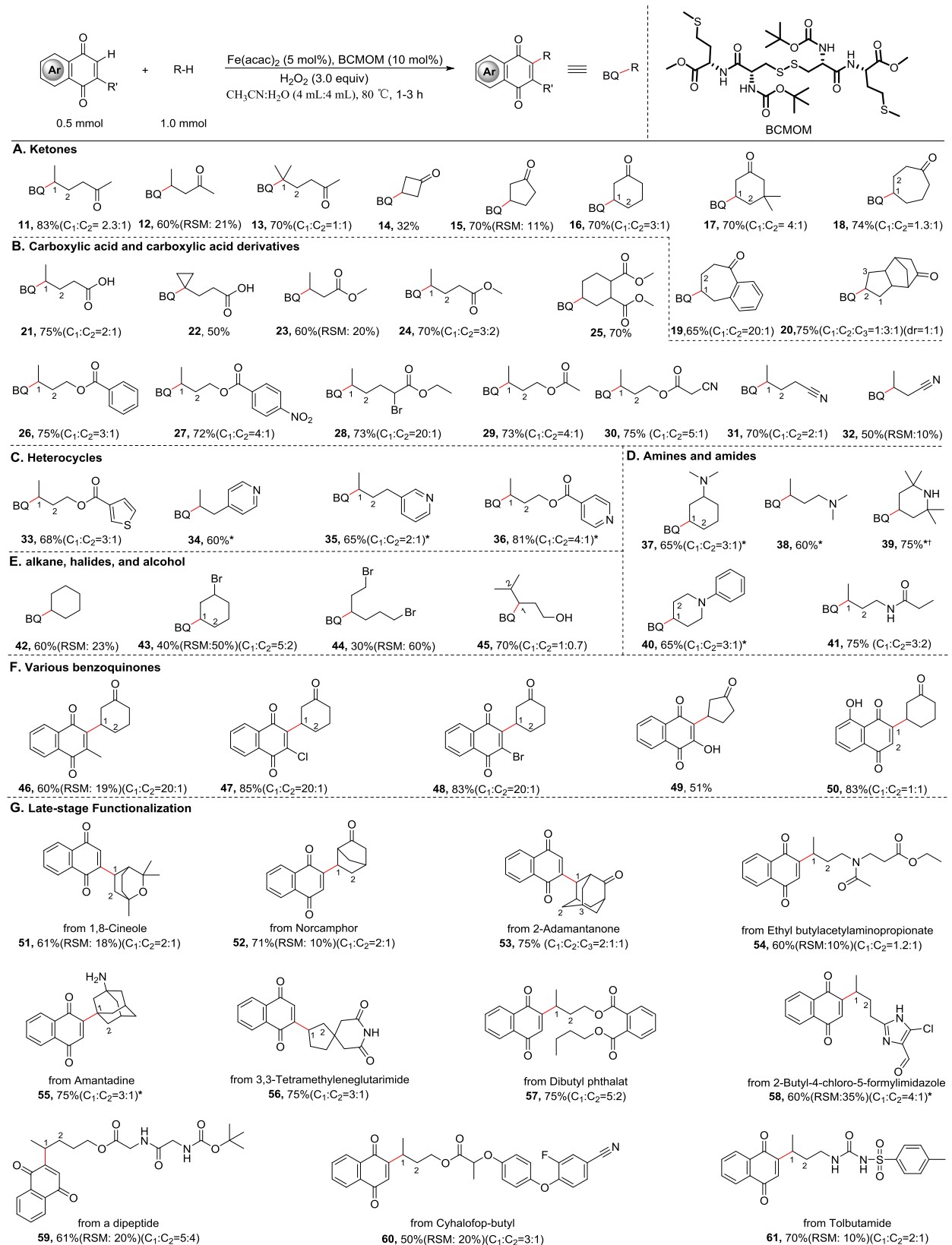

**Fig. 3 | Scope of alkanes in 1,4-quinone C–H functionalization.** Substrates: **A** ketones, **B** carboxylic acid and derivatives, **C** heterocycles, **D** amines and amides, **E** alkane, halides, and alcohol, **F** various benzoquinones, and **G** late-stage functionalization. RSM: recovered starting material. Regioselectivities were determined by $^1$H NMR; * Pre-protonation with $H_2SO_4$; † 5 mmol scale.

the exclusive or major product (**39**, 75% yield; **40**, 65% yield with 75% selectivity), due to the strong inductive electron-withdrawing effect of the protonated nitrogen[46]. Notably, the product **39** was produced on a 5-mmol scale. In addition, other cyclic amines such as *N*-methylpiperidine and *N*-phenylazepane also reacted smoothly to give the desired products in good yields (Supplementary Fig. 19, **160, 161**), further demonstrating the generality of the protocol. For *N*-phenylpyrrolidine, the reaction gave a regioisomer resulting from aryl C–H alkylation, which deviates from the trend observed with other amines (Supplementary Fig. 19, **162**). Substrates such as *N*-methylazocane and *N*-phenylazocane resulted in the formation of an unidentified complex mixture, respectively. Functionalization of acyl-protected amine proceeded without interference from over-oxidations that readily occur when using $H_2O_2$ as the oxidant (**41**).

Having demonstrated the high chemoselectivity and reactivity of BCMOM/Fe for remote methylene functionalization of simple molecules, we next aimed to apply this catalytic system to natural products and drug molecules. 1,8-Cineole, which has anti-inflammatory and antinociceptive effects, gave remote functionalized product **51** in 61% yield. Another bicyclic monoterpenoid, Norcamphor also showed a similar result (**52**, 71% yield). 2-Adamantanone that serves as a molecular probe for characterizing the dimensions and physicochemical properties of the substrate-binding pocket in alcohol dehydrogenases, reacted smoothly to provide **53** as a mixture of regioisomers in 75% yield, but of them the major isomer can be isolated as an analytically pure product. Amantadine, a medication used to treat Parkinson's disease and influenza A infection, also containing an adamantane moiety, was effectively functionalized by the BCMOM/Fe /protonation strategy to afford **55** with higher yield and selectivity (75% yield, 75% regioselectivity). Using the same strategy 2-butyl-4-chloro-5-formylimidazole, an intermediate of Losartan, underwent remote functionalization with 60% isolated yield to afford **58**. Additionally, a biodegradable mosquito-repellent ethyl butylacetylaminopropionate (IR3535) was readily cross-coupled with 1,4-naphthoquinone to produce **54** in a useful yield. Dibutyl phthalate (DBP), Cyhalofop-butyl (a post-emergence herbicide), and Tolbutamide (an oral blood-glucose-lowering drug), housing more oxidatively labile aromatic rings, were regioselectively functionalized at the most electron-rich sp3 C–H sites to give **57**, and **60-61** in 50-75% yields with no observed side products from oxidation of the aromatic functionality.

## Cyclization of 1,4-quinones

1,2,3,4-Tetrahydroquinolinequinone and naphthazepinequinone are prevalent structural motifs in natural products and bioactive molecules. Nevertheless, a concise strategy for the synthesis of them remains elusive[48,49]. Building on the above results in BCMOM/Fe-catalyzed $C_{sp2}$–$C_{sp3}$ couplings of strong aliphatic C–H bonds with 1,4-quinones, we envisaged that an alkyl primary or secondary amine may initially undergo amination/oxidative dehydrogenation with 1,4-quinones to give the aminated 1,4-quinones followed by cyclization via internal molecular $C_{sp3}$–$C_{sp2}$ coupling. An aminated 1,4-quinone intermediate can be isolated, suggesting that the process is potentially feasible (Supplementary Fig. 15). The reaction was performed under standard conditions with pre-protonation of the amine using $H_2SO_4$. To our delight, when linear alkylamines were used as the alkane substrates, BCMOM/FeCl$_2$ afforded 1,2,3,4-tetrahydroquinolinequinone and naphthazepinequinone products with preparative yields and excellent selectivity (Fig. 4). The obtained regiochemical outcomes aligned with the anticipated selectivity pattern characteristic of this catalytic manifold, as previously discussed. For instance, dibutylamine coupled with various naphthoquinones bearing methyl, methoxy, nitro, acetyl, and hydroxy substitutes to provide corresponding 1,2,3,4-tetrahydroquinolinequinones in 48-81% yields (**62–67**). The regioisomeric ratios observed for products **65–67** result from the amine nitrogen connecting to different positions of the naphthoquinone

double bond, leading to distinct regioisomers. The reaction of 1,4-anthraquinone also proceeded well, illustrating the ability to construct a larger fused ring (**68**, 68% yield). Other secondary butylamines such as ethylbutylamine, methylbutylamine, and benzylbutylamine afforded 1,2,3,4-tetrahydroquinolinequinones as well (**74–75** and **80**). We also tested *N*-butylaniline as a substrate, but no desired product was obtained under the standard conditions, likely due to the strong electronic effects of the aniline moiety. In addition, N-propylbutan-1-amine reacted smoothly under the standard conditions to give the corresponding annulation product in 69% yield (Supplementary Fig. 19, **163**). Notably, the one carbon shortened analog of secondary propylamines (dipropylamine, methylpropylamine, ethylpropylamine, diisopropylamine, and methylisopropylamine) underwent terminal methyl functionalization with 42–62% yields (**69–73**), due to methylene sites highly electronically deactivated by the proximal electron-withdrawing amino groups and/or steric inaccessibility of tertiary C–H bonds. In addition, secondary amines bearing longer carbon chains reacted like secondary butylamines, each forming a single 1,2,3,4-tetrahydroquinolinequinone product (**76–79**) from reaction at the γ-methylene site, not the most reactive distal position, probably due to facile generation of less-strained six-member ring. Moreover, more challenging primary amines are viable substrates as well, but underwent each reaction at δ site to give the corresponding seven-member-ring-based naphthazepinequinone (**81–85**), likely reflecting γ-methylene site highly electronically deactivated by a strong electron-deficient protonated primary amino group. Specifically, butylamine (**81**) reacted at the terminal methyl group of the alkyl chain; amylamine (**82**) and lauryl amine (**83**) bearing longer alkyl chains still underwent the coupling at δ C–H bonds. Moreover, aliphatic primary amines bearing oxidatively labile benzene moieties proved to be suitable coupling partners (**84–85**). Finally, we were pleased to find that bioactive compounds, amino acid and nitrogen heterocycles having alkyl chains, also underwent reactions to provide the desired single cyclized products (**86–88**), thus significantly expanding the utility of this transformation.

## Alkylation of azines with alkanes

Considering the pharmaceutical significance of pyridine, quinoline and related heteroaromatic scaffolds, coupled with the distinctive distal selectivity conferred by nitrogen protonation, we initially evaluated our reaction system using a structurally diverse panel of pyridine and quinoline derivatives (Fig. 5). Pre-protonation of heteroarenes and amines with $H_2SO_4$ was performed prior to the reaction. The addition of alkyl radicals to pyridines/quinolines faces a key challenge for controlling site selectivity, owing to these molecules bearing two inherently reactive C2 and C4 positions[50]. Current means mainly rely on activating and blocking C2-sites to realize the corresponding C2- and C4-selectivity using transient or covalently linked species at the pyridine/quinoline nitrogen[51,52]. In our case, the unique BCMOM/Fe enables high (>20:1) regioselectivity for the C2 or C4 addition even in the presence of three activated sites (2C2 and 1C4), providing a practical approach to address the challenge of regioselectivity (C2 vs C4). Exploration began with cyclopentanone as the alkylating reagent. A diverse range of pyridines was efficiently converted to the corresponding products in synthetically useful yields with excellent regioselectivity. Pyridine, a challenging substrate which often undergoes overalkylation and shows poor C2/C4 selectivity in classical Minisci-type additions, exhibited excellent reactivity in this system, affording **89** in 60% yield with >20:1 regioselectivity for the C4 position. This high selectivity may arise from the hydrogen bonding interactions between the ligand BCMOM and protonated pyridine under acidic conditions[53], which disfavors C2 alkylation. Benzo[h]quinoline delivering a nitrogen-directed product (**115**) supports the presence of the interactions. A pyridine ring bearing a polar electron-withdrawing substitute (COOMe) at C2 gave the alkylated product at C4 (**90**), while polar electron-withdrawing substitutes (CN,

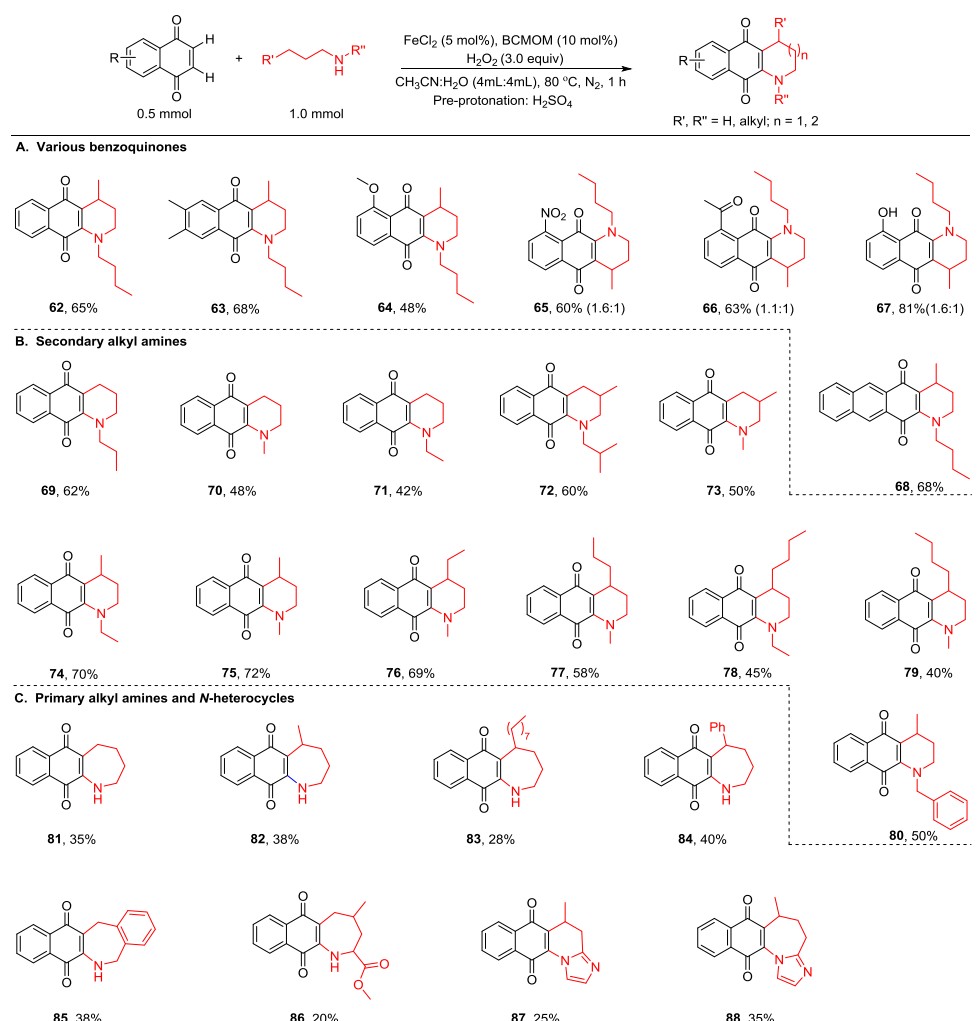

**Fig. 4 | Cyclization of 1,4-quinones with alkanes as alkylating agents. A** Various benzoquinones; **B** secondary alkyl amines; **C** primary alkyl amines and N-heterocycles. The regioselectivities were determined by [1]H NMR. Amine was pre-protonated with $H_2SO_4$.

COOEt, $CF_3$, and COMe) at C3 afforded the corresponding products at C2 (**91**–**94**), suggesting that the noncovalent interactions, primarily coordination and/or hydrogen bonding effects, between BCMOM/Fe and these polar groups overriding between BCMOM/Fe and protonated pyridine nitrogen, thereby blocking the ortho positions of these groups. On the basis of this hypothesis, the high regioselectivity for methyl 5-bromonicotinate is readily comprehensible (**97**). In addition, 3-hydroxypyridine (**95**) and 3-methoxypyridine (**96**) showed similar yields but entirely different regioselectivity, suggesting that strong interactions between the phenolic hydroxy group and BCMOM/Fe that can block its *ortho*-positions [C4 and C2 (left side)], whereas weak interactions between the methoxy and BCMOM/Fe were be overridden by that of the pyridine nitrogen, resulting in steric deactivation of C2 sites. When C4 site was occupied by a substitute, only C2 alkylated product was obtained, as demonstrated by the reactions of methyl isonicotinate and 4-cyanopyridine (**98**–**99**). We then investigated the scope of quinolines (Fig. 5B) and were pleased to discover that under our conditions, excellent regioselectivity (>20:1) was also obtained for unsubstituted and substituted quinolines, following the site-selectivity rules established with the alkylation of pyridines catalyzed by the BCMOM/Fe. Phenyl (**101**), ester (**102**), and chloride (**106**) substituents on the pyridine portion, and methyl (**103**), bromide (**104**), and nitro (**105**) groups on the benzo portion were tolerated, and in all cases very high regioselectivity was observed for addition at a specific position, together with useful yields. The versatility of the current method was

further demonstrated by extending the heteroaromatics scope to pyrimidines, pyridazines, phthalazine, pyrazines, quinoxalines, benzo[h] quinoline, and benzothiazoles (**107**–**118**) (Fig. 5C). In the case of quinoxaline, an alkylation-dehydrogenation product was obtained (**114**, 78% yield).

We next surveyed the scope of various unactivated hydrocarbons to evaluate the generality of this selective C–H heteroarylation (Fig. 5D). The site selectivities can be predicted on the basis of electronic, steric and stereoelectronic environments in a range of carbocyclic and linear alkane structures, as described above. Similar to the reactions of alkanes with naphthoquinones, a series of cyclic and acyclic ketones, including 2-octanone, also gave the corresponding remote C($sp^3$)−H heteroarylation (**119**–**127**, Supplementary Fig. 19, 164), with no heteroarylation of α-C($sp^3$)−H bond observed. Unfunctionalized cyclohexane, which is commonly used as solvent in Minisci-type alkylation owing to its low reactivity, proceeded well (**128**, 58% yield). 7-Oxabicyclo[2.2.1]heptane, an increasingly being utilized in pharmaceutical compound, provided gave only one observed methylene functionalized product **129**. Selective reaction at the most electron-rich, least sterically hindered methylene groups was also observed for various functionalized acyclic alkanes, including those containing nitriles (**130** and **132**), protected /unprotected amines (**133**–**134**). Notably, valeric acid, a typical substrate which can undergo alkylation of heteroarenes by decarboxylation, was compatible and heteroarylated exclusively at the γ position with 68% yield (**131**).

Medicinally relevant compounds were also competent in this process following the site-selectivity rules established from simple compounds. Heteroarylation of norcampher, adamantanone, amantadine, valproic acid, and 2-butyl-4-chloro-5-formylimidazole delivered **135**–**139** in moderate yields with good to excellent regioselectivity, proving additional compatibility with reactive primary amine,

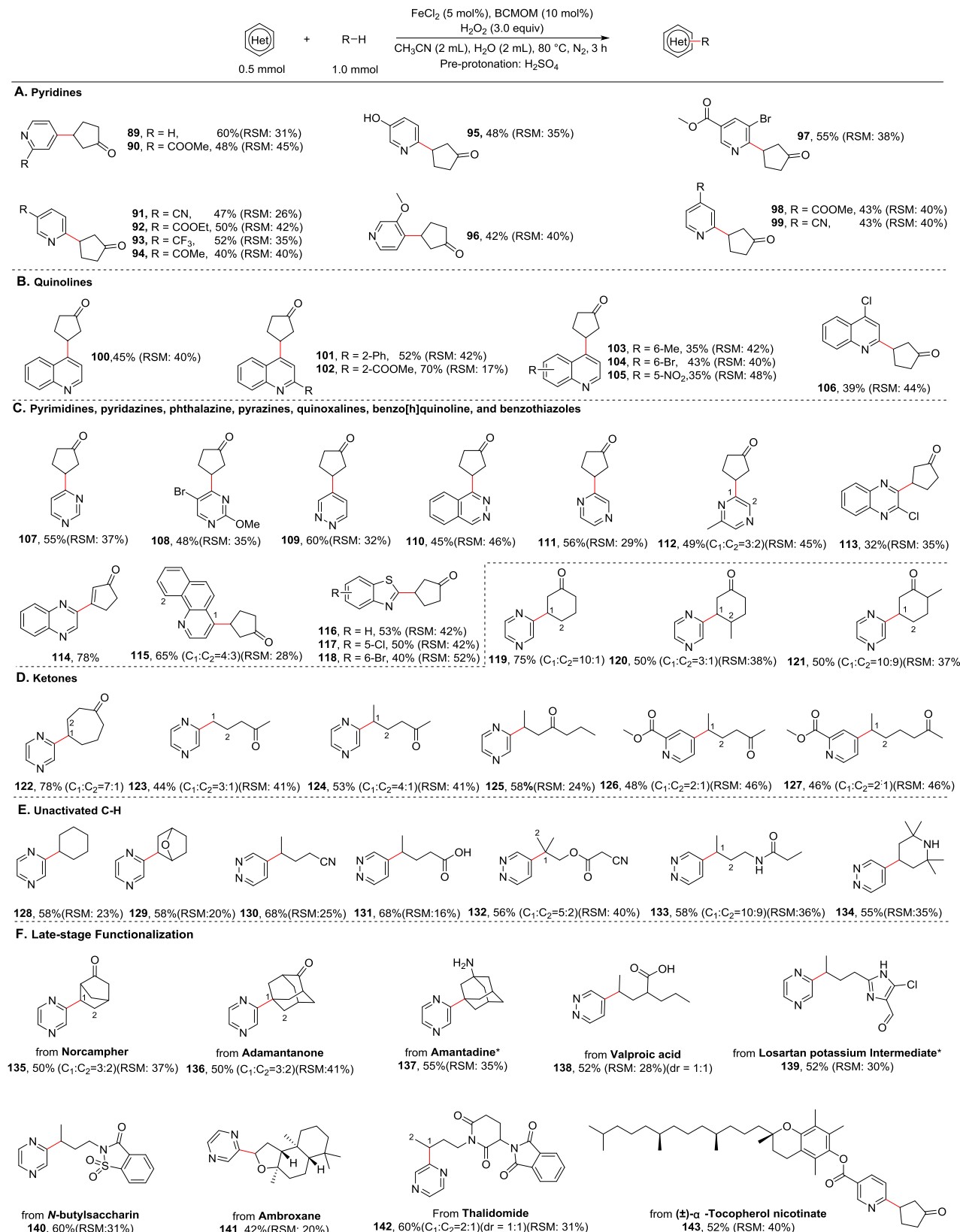

**Fig. 5 | Substrate scope of azine alkylation with alkanes.** Substrates: **A** pyridines, **B** quinolines, **C** other N-heteroaromatics, **D** ketones, **E** unactivated C–H, and **F** late-stage functionalization; Pre-protonation of substrates with $H_2SO_4$.

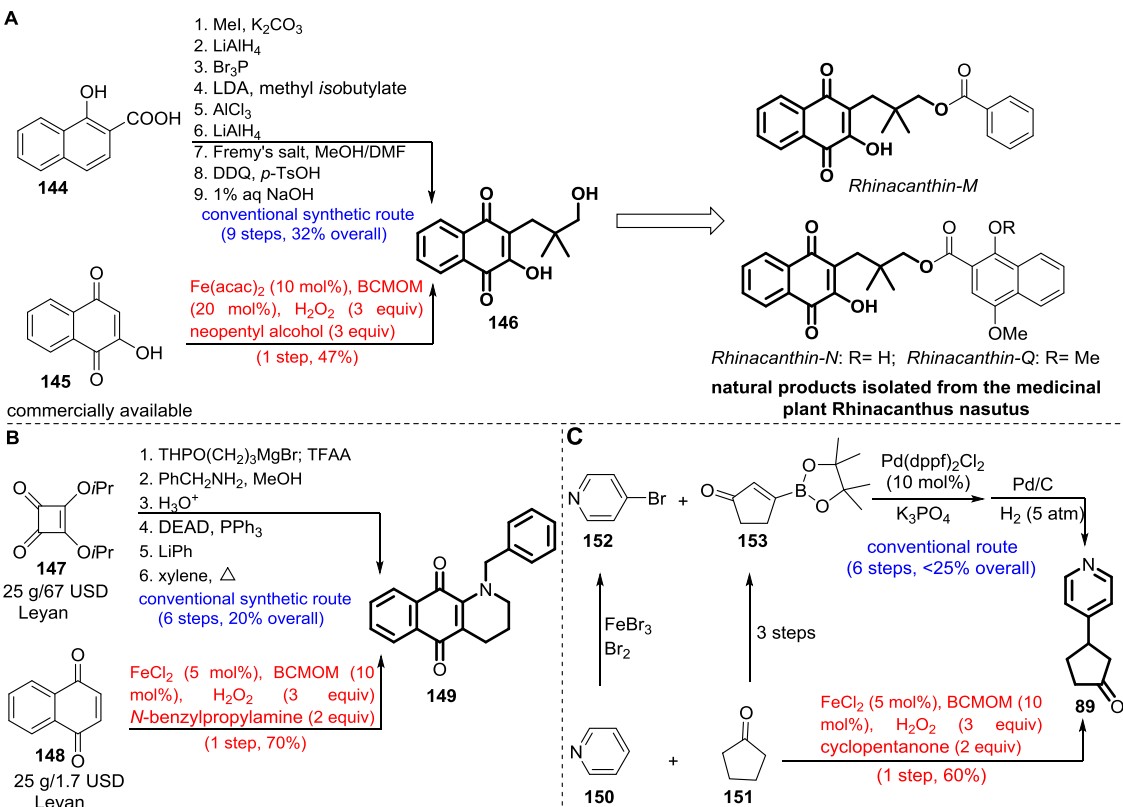

**Fig. 6 | Efficient synthesis of bioactive intermediates via Fe-catalyzed alkylation. A** One-step synthesis of Rhinacanthin-M, -N, and -Q precursor **146** from **145** using Fe(acac)₂/BCMOM, compared to a 9-step conventional route. **B** Synthesis of 1-azaanthraquinone intermediate **149** in one step, versus a 6-step literature route from **147**. **C** Fe-catalyzed one-step synthesis of anti-hepatitis B intermediate **89**, compared with a reported 6-step process. Blue text: conventional routes, Red text: this work.

carboxylic acid and aldehyde functionalities. In addition, *N*-butylsaccharin (skin penetration enhancer) and thalidomide (immunomodulatory drug) were also found to be suitable substrates in this reaction (**140** and **142**). The reaction of a complex terpenoid natural product with a multitude of aliphatic C−H sites—favors the heteroarylated C−H site α to the ether oxygen atom of ambroxide (**141**, 42% yield). Notably, (±)-*α*-Tocopherol nicotinate (a nervous system drug), a pyridine-based complex molecule bearing two C2 and one C4 reactive sites and multiple reactive benzylic positions, effectively and selectively reacted with cyclopentanone to give a single regioisomer **143** in 52% yield, underscoring the great potential of this approach to drug synthesis and construction of complex molecules without the need for costly and laborious de novo synthesis.

## Synthetic applications

To highlight the efficiency of the transformation, we described three representative examples of how this reaction can simplify the synthesis of known important target molecules (Fig. 6). The Rhinacanthin-M, -N and -Q (natural products isolated from the medicinal plant Rhinacanthus nasutus) intermediate **146** has previously been synthesized by a nine-step route involving methylation of acid **144**, reduction, bromination, α-alkylation, lactonization, reduction, oxidation, condensation-oxidation,and hydrolysis-isomerization in 32% overall yield[54]. By contrast, our method allowed the desired product **146** to be prepared directly from readily available **145** in a single step over three hours in 47% isolated yield. Analogously, 1-azaanthraquinone (a pyridoacridine alkaloid) intermediate **149** is known to be accessible from cyclobutenedione **147** in six steps (alkylation, amination, deprotection, Mitsunobu dehydration, arylation, and thermolysis) with 20% yield[55], whereas in our case, direct cyclization of cheaper 1,4-naphthoquinone **148** with

*N*-benzylpropylamine affords the same target **149** in a single operation (70% isolated yield). The anti-hepatitis B virus agent intermediate **89** has previously been prepared from Pd-catalyzed Suzuki coupling of 4-bromopyridine **152** and alkenyl boronate ester **153** followed by Pd-catalyzed hydrogenation[56]. However, both of the prefunctionalized 4-bromopyridine **152** and alkenyl boronate ester **153** generally originated from bromination of raw material pyridine, and a three-step (dehydrogenation, bromination, and borylation) transformation of feedstock cyclopentanone, respectively. Alternatively, the BCMOM/Fe enables direct coupling of the raw materials pyridine and cyclopentanone to deliver the same adduct **89** in 60% yield in only 3 hours. Given the straightforward nature of the modular introduction of carbon moieties into unreactive feedstock alkanes, we believe that the unusual bioinspired iron-catalyzed dehydrogenative C−C coupling will hold great promise for construction of high value-added fine chemicals, as well as benefit future catalyst design for undirected and selective diversification of aliphatic C−H bonds.

## Methods

### General procedure for alkanes as alkylating agents in 1,4-quinone C−H functionalization

A 25 mL flask was charged with Fe(acac)₂ (6.6 mg, 0.025 mmol), BCMOM (37.2 mg, 0.05 mmol), quinone (0.5 mmol), alkane (1.0 mmol) before standard cycles of evacuation and backfilling with dry and pure N₂, and then solvents CH₃CN (4 mL) and H₂O (4 mL) were added. The reaction mixture was stirred under N₂ atmosphere at room temperature for 10 min, and then H₂O₂ (129 μL, 1.5 mmol) was added dropwise into the stirring reaction mixture. Upon completion, the reaction mixture was stirred at 80 °C until no further changes were observed by TLC. The mixture was then allowed to cool to room temperature, and

extracted with ethyl acetate (3 ×10 mL). The organic phases were combined, dried ($Na_2SO_4$), and concentrated to give the crude product. The residue was purified by column chromatography (Petroleum ether/ ethyl acetate) on silica gel to afford the corresponding product.

**General procedure for alkanes as alkylating agents in the cyclization of 1,4-quinones**

A 25 mL flask was charged with $FeCl_2$ (3.2 mg, 0.025 mmol), BCMOM (37.2 mg, 0.05 mmol), quinones (0.5 mmol), and alkyl amine (pre-protonation: 1.0 mmol, mixed with 1.2 mmol $H_2SO_4$) before standard cycles of evacuation and backfilling with dry and pure $N_2$, and then solvents $CH_3CN$ (4 mL) and $H_2O$ (4 mL) were added. The reaction mixture was stirred under a $N_2$ atmosphere at room temperature for 10 min. Then, $H_2O_2$ (129 μL, 1.5 mmol) was added dropwise into the stirring reaction mixture. Upon completion, the reaction mixture was stirred at 80 °C until no further changes were observed by TLC. After the mixture was cooled to room temperature, 2.0 mmol of $Na_2CO_3$ was added, and the resulting mixture was stirred overnight. The mixture was then diluted with a saturated aqueous $Na_2CO_3$ solution (10 mL), and extracted with ethyl acetate (3 ×10 mL). The organic phases were combined, dried ($Na_2SO_4$), and concentrated to give the crude product. The residue was purified by column chromatography (Petroleum ether/ ethyl acetate) on silica gel to afford the corresponding product.

**General procedure for the alkylation of azines with alkanes**

A 25 mL flask was charged with $FeCl_2$ (3.2 mg, 0.025 mmol), BCMOM (37.2 mg, 0.05 mmol), alkane (1.0 mmol), and azine (pre-protonation: 0.5 mmol, mixed with 1.2 mmol con. $H_2SO_4$) before standard cycles of evacuation and backfilling with dry and pure $N_2$, and then solvents $CH_3CN$ (4 mL) and $H_2O$ (4 mL) were added. The reaction mixture was stirred under $N_2$ atmosphere at room temperature for 10 min. Then, $H_2O_2$ (129 μL, 1.5 mmol) was added dropwise into the stirring reaction mixture. Upon completion, the reaction mixture was stirred at 80 °C until no further changes were observed by TLC. The mixture was then allowed to cool to room temperature, diluted with a saturated aqueous $Na_2CO_3$ (10 mL), and extracted with ethyl acetate (3 × 10 mL). The organic phases were combined, dried ($Na_2SO_4$), and concentrated to give the crude product. The residue was purified by column chromatography (Petroleum ether/ ethyl acetate) on silica gel to afford the corresponding product.

## Data availability

All data supporting the findings of this study, including experimental procedures, compound characterization (e.g., NMR spectra), and mechanistic studies, are available within the Article and its Supplementary Information.

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

## Acknowledgements

The work was sponsored by the Natural Science Foundation of China (22371126 and 21776139, W. H.), the Jiangsu Basic Research Center for Synthetic Biology (BK20233003, W.H.), the "Qing Lan Project" Young and Middle-aged Academic Leaders of Jiangsu Provincial Colleges and Universities (W.H.), the Natural Science Foundation of Jiangsu Province (BK20161553, W.H.), and the Priority Academic Program Development of Jiangsu Higher Education Institutions (W.H.).

## Author contributions

L. Z., H. C., D. X., K. S., S. Zhu and M. G. performed and analyzed the experiments. W. H. conceived and supervised the project. L. Z., H. C., D. X., K. S. and W. H. wrote the manuscript.

## Competing interests

The authors declare no competing interests.
