## [Transparent Peer Review file · Nature Communications]

Iron-catalyzed aliphatic C–H functionalization to construct carbon–carbon bonds

Corresponding Author: Professor Wei Han

Version 0:

Reviewer comments:

Reviewer #1

(Remarks to the Author)

Recommendation: Publish in Nature Communication after minor revisions.

This manuscript reported by Han and coworkers disclosed a noncanonical bioinspired ligand BCMOM-enhanced iron-catalyzed strong aliphatic C–H functionalization with 1,4-quinones and azines. Preliminary mechanistic studies and experimental results suggest the involvement of electrophilic iron-oxo species process, allowing for the use of diverse alkanes as limiting reagent reacting efficiently and the site selectivity predicted on the basis of steric, electronic, and stereoelectronic environments even in complex molecular settings. In classic bioinspired iron catalysis, iron-oxo species readily transfer an oxygen atom to the substrate through a hydrogen abstraction/oxygen recombination pathway known as the oxygen rebound mechanism. It was reasonable to believe that any transformations that involved iron-oxo intermediates would necessarily lead to the formation of oxygenated products. In fact, a long-standing challenge for implementing the radical rebound approach for catalytic aliphatic C–H functionalization has been the suppression of the oxygenation products generated via oxygen rebound, especially for intermolecular carbon–carbon bond formation reactions. This work takes the unconventional bioinspired catalysis approach to achieve the first breakthrough of a thiolate-ligated iron–oxo favoring hydrogen-atom abstraction over oxygen-atom transfer to construct carbon–carbon bonds, thus holding great promise for wide-ranging and efficient diversification of feedstock alkanes. Therefore, I rate the quality of the manuscript, that is written well, as very high and support publication in Nature Communications subject to minor modifications:

1. Considering that the BCMOM is a chiral ligand, it is interesting to know if it is possible to run the current transformation in an enantioselective way.
2. In Fig. 3, a series of aliphatic amines delivered the desired products in synthetic useful yields. Did the authors try to use an N-alkylaniline as substrate? What is the result?
3. Sequential numbers of the references should not appear in the abstract.

Reviewer #2

(Remarks to the Author)

In this paper, with in situ generated iron catalyst by mixing a simple iron salt with a ligand, Han and co-workers reported an iron catalyzed crossing coupling reactions to construct new C–C bonds. Although the conformation of the iron catalyst and the active iron species are not clear, the reaction shows good performance by giving near 150 examples in moderate to good product yields without oxygenation or desaturation, and with several examples in >90% product yields when calculated based on RSM. To generate carbon radicals from alkanes via oxidation reactions for cross coupling reactions remains a big challenge. In many cases, it means harsh reaction conditions with low desired product yields. However, in this paper, Han and co-workers showed the good performance of their catalytic system for such reactions. Although impressive results have been presented, there are some questions need to be well addressed before the acceptance for publishing in Nature Communications.

The authors emphasized the importance of using limiting amounts of alkanes as substrates (lines 39, 42 and 77) and claimed that they conducted the reactions with alkanes as limiting reagents (lines 18, 96 and 391). However, 2 equiv. of alkanes were used in their work. So, how to define “limiting”? If “2 equiv.” can be looked as “limiting”, why “3 equiv.” cannot be looked as “limiting”? Why “4 or 5 equiv.” cannot be looked as “limiting”?

Lines 100-102, the ¹⁸O incorporation experiment just revealed that a small amount of ¹⁸O was incorporated into the product. However, it is not determinative evidence for “supporting that the active oxidant is a high-valent oxoiron”. In addition, Ref 37 is a report on chromium(V)-oxo species, which has a low correlation with what the authors claimed “often oxoiron(IV)”.

Line 105, KIE experiment. The KIE values should be determined via both parallel/individual and competitive experiments. In addition, The KIE values should be obtained by measuring the reaction rates for both proton- and deuterated substrates, not just by comparing the product yields.

Line 113, Fig. 1D. It is very unreasonable to propose such a complex reaction mechanism without any experimental supports.

Reaction optimization. How to determine the best Fe/Ligand ratio is 1:2? How to determine the best Fe loading is 5 mol%? How to determine the best amounts of H₂O₂ are 3 equiv.? How to determine the best reaction temperature is 80 °C? How to determine which Fe salt to use in the reaction, especially for the substrates pre-protonated with H₂SO₄?

Lines 125-126, More than 2-hexanone, 2-heptanone and 2-octanone should be added as examples to confirm “favoring selective functionalization at the most remote and least electronically deactivated site”.

Lines 134-139, To compare regioselectivity with products 18 and 19, eight-membered ketones should be added as examples.

Line 134, Compounds 20 cannot be looked as a nine-membered ketone.

Lines 148, more than valeric acid, n-hexanoic acid, n-heptanoic acid and n-octanoic acid should be added as examples.

Line 149, the “cyclopropyl radical” mentioned by the authors is stable and difficult to undergo ring-opening reaction. In addition, 4-cyclopropylbutanoic acid should be added for comparison.

Line 185, more than using a bulky tetramethylpiperidine as substrate (39), cyclic amines including N-methylpiperidine, N-methylazepane, N-methylazocane, N-phenylpyrrolidine, N-phenylazepane and N-phenylazocane should be added as examples.

Lines 187-189, It is difficult to understand “Functionalization of acyl-protected piperidine proceeded without interference from over-oxidations that readily occur when using H₂O₂ as the oxidant (41)”. In addition, why an acyl amide needs to be pre-protonated.

Fig. 3, the reactions in this figure are 2-step, one-pot reactions, however, most of the reactions gave products in low to moderate yields. Comparing with Fig. 2, why intramolecular reactions gave lower product yields than intermolecular reactions? Are there any aminated 1,4-quinone intermediates left? Interestingly, the authors still used 3 equiv. of H₂O₂ as oxidant, without taking the amount for oxidative dehydrogenation into account. So, what will happen if the reaction is conducted with more equiv. of H₂O₂? How many equiv. of H₂O₂ are needed if the reaction starting from aminated 1,4-quinone intermediate (lines 217-220)?

Lines 231, comparing with product 74, what product will be obtained with N-propylbutan-1-amine as a substrate?

Line 249, It is interesting to find that single cyclized products 87 and 88 were obtained. Why there is no 7-membered ring product for 87, and no 6-membered ring product for 88?

Both in the main text and in Fig. 4, the authors should clearly figure out that all azines were used as H₂SO₄ salts. In Fig. 4, the author just made a * annotation to 137 and 139, which is far from the reaction conditions described in SI.

Lines 267, 273 and 279, the authors used many “presumably” to explain their results. It is difficult to understand these hypotheses. As the authors claimed in lines 171-176, the basic N was protonated because “the pyridine and amine functionality may coordinate to and inhibit the catalyst or undergo undesirable side reactions”, However, in line 268, the authors explained “interactions between the bulky BCMOM/Fe and the pyridine nitrogen”. It seems that the authors forgot that the pyridine had been protonated.

Products 126 and 127 were obtained in similar product yields (48% and 46%) with same RSM and C1:C2 ratio. What will happen if use 2-octanone as a substrate?

Minor points:

Lines 21-22. “overcome the inherent problem of the P450s”. Why it is a “inherent problem”? Both in Nature and in laboratories, P450s are not used to generate radicals for cross coupling.

Line 48, Fig. 1A. Is it “O₂ + 2H⁺”?

Please change the item “oxoiron” to “iron-oxo”.

The sentence in line 59-64 “Implementing high-valent iron-oxo as the catalytic species for efficient alkane functionalization while simultaneously avoiding oxygenation, remains an important challenge, particularly in intermolecular carbon-carbon bond formation reactions(Fig. 1B) because unlike heteroatoms carbon is difficult to coordinate with a transition-metal catalyst and easy to be oxygenated under oxidation conditions even if a carbon-metal species can be formed.” is too long.

Line 73: require the the substrate

Figure 2. Please also show the amounts of BQ and alkanes in the reaction scheme. In addition, for the reactions with < 75% product yields, please show the RSM.

Line 134: afforded

Line 143: 46-48?

Line 157: menthyl?

Comparing with products 46-48, was compound 50 obtained with exclusive β-site regioselectivity?

Line 221: used as the the alkane substrates

Fig. 3, please show the amounts of BQ, amines and solvent in the reaction scheme. In addition, for the reaction with < 75% product yields, please show the RSM.

Line 266: is an excellent substrate

Fig. 4, please show the amounts of BQ and amines in the reaction scheme. In addition, for the reaction with < 75% product yields, please show the RSM.

Line 271: electron-withdrawing substitute (COOMe) at C3, it should be C2?

Figure 5, what is the reason to use different iron salts with different loading amounts?

Although it is not mandatory, it is surprising to find a paper that does not have discussion and conclusion sections. References: please unify the citation format of references. Some references showed all authors, whereas some shortened the list of authors by et al.

SI.

The authors need to check the amounts of reagents they used in the reaction. For examples, line 54, H₂O₂ (1.5 mmol, 64 μ L); line 108, H₂O₂(22 μ L, 0.375mmol); line 123, H₂O₂ (35%) (129 μ L, 1.5 mmol). line 50, BCMOM(0.025mmol, 18.6mg); line 107, BCMOM (9.3 mg, 0.025 mmol). line 121, H₂SO₄(67.5 μ L, 2.4 equiv.), line 1841, H₂SO₄(66.0 μ L, 2.4 equiv.).

Table S1, entry 6, there is no chemical structure for (BOC-CYS-OH)₂. There is no catalytic result for BCSOM.

Lines 145-146. The reaction conditions (Fe loading, 10 mol%) did not consist with Table S1 (5 mol%).

Lines 152, 214, 228, 243, and 258, the authors wrote "until the reaction was completed (observed by TLC)". In most cases, the reagent BQ did not have a 100% conversion, how to determine the completion of reaction by TLC?

I checked the characterization data and NMR spectra of compounds 13-20, I think the authors need to pay more attention to assignment of peak-splitting and coupling constants. For examples, Line 276: 1.61 (dd, J = 14.6, 7.3 Hz; Line 319: 2.86 (qd, J = 17.9, 7.4 Hz; Line 334, 8.11 (ddd, J = 8.8, 5.6, 3.6 Hz; Line 417, 1.97 (dd, J = 14.3, 3.1 Hz), 1.68-1.47 (dd, J = 23.6, 11.7 Hz; Line 434, 8.17-8.05 (m; Line 436, 3.10 (ddd, J = 21.2, 14.6, 6.2 Hz; line 455, 6.85 (s, 1 H, minor), same ratio, no one is minor. Line 455, 2.99 (dt, J = 13.7, 5.6 Hz); Line 456, 2.55 (s, 1 H, minor), 2.48 (s, 1 H), 2.44 (s, 1 H), 2.39 (dd, J = 17.9, 9.0 Hz, 2 H); Line 457, 1.76 (ddd, J = 17.3, 15.4, 5.3 Hz; Line 458, 1.35 (ddd, J = 28.5, 13.2, 6.7 Hz,. The authors need to carefully re-check all characterization data and spectra in SI.

The characterization data and spectra of product 13-2 was missing. If the minor products were obtained in calcd. > 15% product yields (such as overall 60% product yield with 1:3 ratio or overall 70% product yield with 1:4 ratio), the characterization data and spectra of them need to be provided. Please also check the characterization data report on other minor products in this work.

The ¹³C NMR of compound 16-1 is not pure. In addition, in many spectra, there is an unassigned signal around 3.70 ppm (such as in page S170, S174, S176, S178, S184, S200) in ¹H NMR and an unassigned signal around 58 ppm (such as S161, S171, S173, S179, S187, S189, S195, S201) in ¹³C NMR. Are these signals correlated to impurity or side products?

The amount of H₂SO₄ used in preparation of compounds 89-143 is elusive. In the preparation of pyridine derivatives that contain one basic N, 2.4 equiv. of H₂SO₄ was used; however, in the preparation of pyrimidines, pyridazines and others that contain two basic N, 2.4 equiv. of H₂SO₄ was still used; furthermore, even in preparation of compound 139 that contain three basic N, 2.4 equiv. of H₂SO₄ remained unchanged. In addition, 1.2 equiv. of H₂SO₄ was used to protonate one basic N in Fig. 2 and Fig. 3, why 2.4 equiv. of H₂SO₄ was used here to protonate one basic N?

Throughout the SI, "unknow compound" was shown for many times. What is it?

Reviewer #3

(Remarks to the Author)

The manuscript by Wei Han group reports an efficient protocol for the iron-catalyzed arylation of C-H bonds with benzoquinones and azines. This novel methodology uses Fe/H₂O₂ chemistry to abstract hydrogen atoms from aliphatic C-H bonds and generate alkyl radicals, which are then trapped by benzoquinones or protonated azines. The authors propose (Figure 1) a cyt P450 like mechanism involving a key high-valent iron-oxo species as the C-H abstractor, with the typical oxygen-rebound prevented by the thiolate ligand. The products are obtained in good yields, the reaction has a broad scope and application to pharmaceutically relevant molecules is shown. The SI are well-organized and the compounds are adequately characterized.

Overall, the synthetic results are convincing, and the protocol represents an interesting and new methodology, with promising synthetic potential. Moreover, the design of the system is interesting. However, in the proposed mechanism part there are some points that should be clarified. Therefore, I support publication of this work in Nat Commun after addressing these points.

The effect of the thiolate ligand in "implementing high-valent iron-oxo as the catalytic species for efficient alkane functionalization while simultaneously avoiding oxygenation" is a key point for catalyst design that, in my opinion, deserves further elucidation.

First, part of the optimization results in Table S1, especially those showcasing the role of the thiolate ligand, should be moved to the main text.

Then, limited evidence for the involvement of a high-valent iron-oxo species is provided. The main one is an indirect experiment previously reported (Science 2021, 374, 77) in which 2.4% of ¹⁸O labelled arene oxygenation product is detected in the presence of ¹⁸O labelled water. Given the low incorporation, the related blank experiments (in normal water and exposure of the product to labelled water) as well as the experimental errors need to be added to support the data.

Additional spectroscopic or MS evidence for an iron-oxo or another intermediate in Figure 1d would be needed to confirm their involvement and support the proposed mechanistic scheme. Otherwise, other pathways than those shown in Figure 1d cannot be convincingly ruled out and should be considered and discussed in the manuscript. For instance, Fe salts and peroxides are known to generate electrophilic oxyl radicals (Fenton reaction) competent for C-H abstraction with a selectivity that is often similar to iron-oxos (see for instance Coord Chem Rev 2000, 200, 517 or J Biol Inorg Chem 2017, 22, 425).

I am also curious about the lack of oxygenated products reported. Is there any trace of oxygenated byproducts in the reaction mixture? Even in the absence of the arene trap?

No information on the actual structure of the iron catalyst in solution is provided. What is the stoichiometry of the complex? Is there evidence for the proposed key thiolate-Fe bond in solution? And is such function retained under oxidative conditions?

Minor points:

- Page 4: The discussion of functionalization site would be clearer by naming proximal and distal positions to the functional group instead of β , γ or δ .
- Page 5, lines 163-167. Why is there no α -functionalization of the alcohol at all?
- Page 5, line 187: why is product 41 discussed as an acyl protected piperidine?
- Page 6, line 218-220: add the related experiment to the main text and figures. Also the pre-protonation conditions need to be explicitly stated in the text and the figure.
- Page 7, line 271: I think it refers to C2 position
- Page 7, lines 268 and 274: which are the "noncovalent interactions" mentioned?
- Figures: the pre-protonation conditions should be made clearer at first sight
- Figure 2, product 19: Why isn't any benzylic functionalization observed? Product 20: add the d.r.
- Figure 3, products 65-67: what the reported ratios stand for needs to be explicitly stated
- Figure 4: with the same alkane substrate (i.e. 119 and 16, 122 and 18, 124 and 11), one expects the same site-selectivity for hydrogen abstraction to be determined by the reactivity of the iron-oxo. Why does the selectivity change upon changing the arene partner?

Version 1:

Reviewer comments:

Reviewer #1

(Remarks to the Author)

The authors have addressed all the issues from the reviewers. The paper can be accepted in its current form.

Reviewer #2

(Remarks to the Author)

The authors have addressed the points raised by the reviewers. The article can be accepted in the current version. The catalytic data described in this paper are good, however, mechanistic studies are still weak. The authors may want to devote more effort to providing insights into the possible reaction mechanisms in their future work.

Reviewer #3

(Remarks to the Author)

The authors did a thorough work and appropriately addressed the points raised. The manuscript is now suitable for publication.

Dear Editors and Reviewers:

Thank you for your letter and for the reviewers' comments concerning our manuscript entitled "**Iron-catalyzed aliphatic C–H functionalization to construct carbon–carbon bonds**" (Manuscript Number: NCOMMS-24-76820-T). Those comments are all valuable and very helpful for revising and improving our paper, as well as the important guiding significance to our research. We have studied comments carefully and have made correction and hope to meet with approval. The revised section is highlighted in the paper with a yellow background. The responds to the reviewer's comments are as follow:

Reviewer(s)' Comments to Author:

Reviewer: 1

Comments:

Recommendation: Publish in Nature Communication after minor revisions.

This manuscript reported by Han and coworkers disclosed a noncanonical bioinspired ligand BCMOM-enhanced iron-catalyzed strong aliphatic C–H functionalization with 1,4-quinones and azines. Preliminary mechanistic studies and experimental results suggest the involvement of electrophilic iron-oxo species process, allowing for the use of diverse alkanes as limiting reagent reacting efficiently and the site selectivity predicted on the basis of steric, electronic, and stereoelectronic environments even in complex molecular settings. In classic bioinspired iron catalysis, iron-oxo species readily transfer an oxygen atom to the substrate through a hydrogen abstraction/oxygen recombination pathway known as the oxygen rebound mechanism. It was reasonable to believe that any transformations that involved iron-oxo intermediates would necessarily lead to the formation of oxygenated products. In fact, a long-standing challenge for implementing the radical rebound approach for catalytic aliphatic C–H functionalization has been the suppression of the oxygenation products generated via oxygen rebound, especially for intermolecular carbon–carbon bond formation reactions. This work takes the unconventional bioinspired catalysis approach to achieve the first breakthrough of a thiolate-ligated iron–oxo favoring hydrogen-atom abstraction over oxygen-atom transfer to construct carbon–carbon bonds, thus holding great promise for wide-ranging and efficient diversification of feedstock alkanes. Therefore, I rate the quality of the manuscript, that is written well, as very high and support publication in Nature Communications subject to minor modifications:

We would like to thank the reviewer for the helpful comments and suggestions. We have carefully addressed these points and revised the manuscript accordingly. Below, we provide our point-by-point responses.

(1) Considering that the BCMOM is a chiral ligand, it is interesting to know if it is possible to run the current transformation in an enantioselective way.

We thank the reviewer for this insightful comment. Although BCMOM is chiral, the reaction proceeds via a radical pathway, where enantioselectivity is challenging to control due to rapid radical recombination. Under the reported conditions, the chiral environment of BCMOM does not induce stereoselectivity. Future studies may explore strategies to achieve enantioselective control.

(2) In Fig. 3, a series of aliphatic amines delivered the desired products in synthetic useful yields. Did the authors try to use an N-alkylaniline as substrate? What is the result?

We appreciate the reviewer's question. We did attempt the reaction with N-butylaniline as a substrate; however, no desired product was observed under the standard conditions. We speculate that the strong electronic effects of the aniline moiety may inhibit the reaction.

(3) Sequential numbers of the references should not appear in the abstract.

We thank the reviewer for pointing this out. In accordance with the journal's guidelines, we have removed the reference numbers from the abstract. The revised abstract no longer includes any citation numbers.

Reviewer: 2

Comments:

In this paper, with in situ generated iron catalyst by mixing a simple iron salt with a ligand, Han and co-workers reported an iron catalyzed crossing coupling reactions to construct new C-C bonds. Although the conformation of the iron catalyst and the active iron species are not clear, the reaction shows good performance by giving near 150 examples in moderate to good product yields without oxygenation or desaturation, and with several examples in >90% product yields when calculated based on RSM. To generate carbon radicals from alkanes via oxidation reactions for cross coupling reactions remains a big challenge. In many cases, it means harsh reaction conditions with low desired product yields. However, in this paper, Han and co-workers showed the good performance of their catalytic system for such reactions. Although impressive results have been presented, there are some questions need to be well addressed before the acceptance for publishing in Nature Communications.

We thank the reviewer for the thoughtful summary of our work and for recognizing its significance. Below, we address the specific questions raised and have revised the manuscript accordingly.

(1) The authors emphasized the importance of using limiting amounts of alkanes as substrates (lines 39, 42 and 77) and claimed that they conducted the reactions with alkanes as limiting reagents (lines 18, 96 and 391). However, 2 equiv. of alkanes were used in their work. So, how to define “limiting”? If “2 equiv.” can be looked as “limiting”, why “3 equiv.” cannot be looked as “limiting”? Why “4 or 5 equiv.” cannot be looked as “limiting”?

We appreciate the reviewer’s comment. To clarify, we have replaced the term “limiting reagent” with “substrate in minimal excess” throughout the manuscript, as our conditions involve using only 2 equiv. of alkanes. This represents a substantial improvement over conventional approaches, where alkane substrates are often used in large excess (>5 equivalents) or even as solvents. By significantly reducing the alkane loading, our method offers a more practical and efficient approach to aliphatic C-H functionalization.

(2) Lines 100-102, the ^{18}O incorporation experiment just revealed that a small amount of ^{18}O was incorporated into the product. However, it is not determinative evidence for “supporting that the active oxidant is a high-valent oxoiron”. In addition, Ref 37 is a report on chromium(V)-oxo species, which has a low correlation with what the authors claimed “often oxoiron(IV)”.

We thank the reviewer for this important comment. The reason for the small amount of ^{18}O incorporation is in that the rate of oxygen exchange between high-valent iron oxo intermediate and H_2^{18}O is much slower than that of oxygen transfer from the intermediate to the arene substrate (C-H cleavage of arene was not involved in the rate determining step)^{S2}, as demonstrated in previous studies^{S3,S4}. This result suggests the formation of oxoiron under normal reaction conditions (included in the Supplementary Information). Furthermore, an ESI mass-spectrometry (MS) analysis of the model reaction mixture revealed a prominent peak at $m/z = 1731.4928$, consistent with the chemical structure of $[(\text{acac})_2(\text{BCMOM})_2]^+ \text{Fe}^{\text{IV}}(\text{O}) + \text{H}]^+$ high-valent oxoiron species (included in the Supplementary Information). The corresponding discussion has also been added to the revised manuscript. Additionally, we replaced Ref. 37 with a more suitable reference to support our proposed iron-oxo species mechanism.

(3) Line 105, KIE experiment. The KIE values should be determined via both parallel/individual and competitive experiments. In addition, The KIE values should be obtained by measuring the reaction rates for both proton- and deuterated substrates, not just by comparing the product yields.

We appreciate the reviewer’s comment. The KIE value has been obtained by measuring the reaction rates for both proton- and deuterated substrates and has been added in the ESI.

(4) Line 113, Fig. 1D. It is very unreasonable to propose such a complex reaction mechanism without any experimental supports.

We thank the reviewer's comment. Although detailed mechanistic studies have not yet been conducted, several control experiments reveal the general features of the mechanism. First, when the BCMOM/Fe catalyst was used for oxidation of arene C-H (a specific acetanilide) with H₂O₂ in CH₃CN/H₂O¹⁸, O¹⁸ incorporation into the desired product was observed, implying the formation of oxoiron under normal reaction conditions. Moreover, HRMS analysis of iron species are consistent with the chemical structure of [(acac)₂(BCMOM)₂⁺Fe^{IV}(O)] (at m/z = 1731.4928 ([M+H]⁺)) species (included in the Supplementary Information). These results suggest the presence of oxoiron under normal reaction conditions. In addition, the reaction fails to proceed in the absence of H₂O (see the Supplementary Information), consistent with cytochrome P450 catalysis mechanisms. This observation suggests that water molecules likely serve as axial ligands and may participate in the proton-transfer cascade that facilitates heterolytic O-O bond cleavage (see also: *Biochemistry* 2001, 40, 45, 13456–13465). Furthermore, hydrogen atom transfer (HAT) step provides alkyl radical intermediates that can be detected by radical scavenger BrCCl₃. According to the above studies, a plausible mechanism for the iron-catalyzed aliphatic C-H functionalization is suggested, as outlined in Fig. 1D.

(5) Reaction optimization. How to determine the best Fe/Ligand ratio is 1:2? How to determine the best Fe loading is 5 mol%? How to determine the best amounts of H₂O₂ are 3 equiv.? How to determine the best reaction temperature is 80 °C? How to determine which Fe salt to use in the reaction, especially for the substrates pre-protonated with H₂SO₄?

We appreciate the reviewer's comment. We have further optimized the Fe/Ligand ratio, Fe loading, amounts of H₂O₂, and reaction temperature, and indicate that these parameters are optimal (see Table S1 of the Supplementary Information). As described in Table S1, Fe(acac)₂ and FeCl₂ afforded comparable excellent results. For alkylation of azines and cyclization of 1,4-quinones, FeCl₂ yielded slight better results than Fe(acac)₂.

(6) Lines 125-126, More than 2-hexanone, 2-heptanone and 2-octanone should be added as examples to confirm "favoring selective functionalization at the most remote and least electronically deactivated site".

We thank the reviewer for the helpful suggestion. We have now investigated 2-heptanone and 2-octanone. These examples also indicated "favoring selective functionalization at the most remote and least electronically deactivated site". However, the number of regioisomers increases with longer carbon chains (as described in a previous study, see also: *Science*, 2010, 327, 566-571), thus leading to produce intractable isomeric mixtures. The results, including regioisomeric ratios and total yields, are presented in the Supplementary Information (154, 155), and the relevant discussion has been added in the main text.

(7) Lines 134-139, To compare regioselectivity with products 18 and 19, eight-membered ketones should be added as examples.

We thank the reviewer for the suggestion. We have now included the eight-membered ketone example (see SI, **156**) and added the corresponding discussion in the main text to support the observed regioselectivity trend.

(8) Line 134, Compounds 20 cannot be looked as a nine-membered ketone.

We appreciate the reviewer's careful observation. We acknowledge that Compound **20** was misclassified as a nine-membered ketone. To ensure accuracy, we have revised the text to describe it as a polycyclic ketone in the revised manuscript.

(9) Lines 148, more than valeric acid, n-hexanoic acid, n-heptanoic acid and n-octanoic acid should be added as examples.

We thank the reviewer for the helpful suggestion. Additional examples with n-hexanoic acid, n-heptanoic acid, and n-octanoic acid have now been included in the Supplementary Information (see SI, **157-159**), and the relevant discussion has been added to the main text.

(10) Line 149, the "cyclopropyl radical" mentioned by the authors is stable and difficult to undergo ring-opening reaction. In addition, 4-cyclopropylbutanoic acid should be added for comparison.

We thank the reviewer for the valuable suggestion. We have attempted the reaction of 4-cyclopropylbutanoic acid under our standard conditions. However, the reaction gave low conversion and no major product could be isolated. These results suggest that this substrate may not be compatible with the current catalytic system, possibly due to competing side reactions or decomposition. We have therefore not included this example in the main text but will consider further investigation in future studies.

(11) Line 185, more than using a bulky tetramethylpiperidine as substrate (39), cyclic amines including N-methylpiperidine, N-methylazepane, N-methylazocane, N-phenylpyrrolidine, N-phenylazepane and N-phenylazocane should be added as examples.

We appreciate the reviewer's valuable suggestion. In response, we have tested several representative cyclic amines under the standard reaction conditions. N-Methylpiperidine and N-phenylazepane reacted smoothly to afford the desired products in good yields (see SI, **160, 161**). N-Methylazepane was also examined, but no major product was detected under the current conditions. For N-phenylpyrrolidine, the reaction gave a regioisomer resulting from aryl C-H alkylation, which deviates from the trend observed with other amines (SI, **162**). Substrates such as N-methylazocane and N-phenylazocane resulted in the formation of an unidentified complex mixture. These results collectively demonstrate the applicability and scope of our methodology across diverse cyclic amine structures.

(12) Lines 187-189, It is difficult to understand "Functionalization of acyl-protected piperidine proceeded without interference from over-oxidations that readily occur when using H₂O₂ as the

oxidant (41)". In addition, why an acyl amide needs to be pre-protonated.

We thank the reviewer for the insightful comment. We have corrected the description in the main text by replacing "acyl-protected piperidine" with "acyl-protected amine" to accurately reflect the substrate used in compound **41**. For the reaction of the acyl-protected amine, we run a control experiment, thus indicating that pre-protonation is really not necessary.

(13) Fig. 3, the reactions in this figure are 2-step, one-pot reactions, however, most of the reactions gave products in low to moderate yields. Comparing with Fig. 2, why intramolecular reactions gave lower product yields than intermolecular reactions? Are there any aminated 1,4-quinone intermediates left? Interestingly, the authors still used 3 equiv. of H₂O₂ as oxidant, without taking the amount for oxidative dehydrogenation into account. So, what will happen if the reaction is conducted with more equiv. of H₂O₂? How many equiv. of H₂O₂ are needed if the reaction starting from aminated 1,4-quinone intermediate (lines 217-220)?

We thank the reviewer for the insightful comment. In Fig. 4, the in situ generated aminated 1,4-quinone intermediates became electron-rich, while alkyl radical is nucleophilic, thus leading to lower the reactivity of the intramolecular alkylation of 2-aminoquinones. It can be supported by the result from the reaction of a 2-hydroxyquinone (**49** in Fig 2). In the case of lower product yields, aminated 1,4-quinone intermediates can be observed, albeit in little amount. And the aminated 1,4-quinone intermediates can be oxidized into the corresponding unidentified side products. When the reaction is conducted with more equiv. of H₂O₂ (4-5 equiv.), the reaction cannot be further improved. Two equiv. of H₂O₂ are enough if the reaction starting from aminated 1,4-quinone intermediate. More equiv. of H₂O₂ can cause the problem of chemo selectivity.

(14) Lines 231, comparing with product **74**, what product will be obtained with N-propylbutan-1-amine as a substrate?

We thank the reviewer for the insightful question. We have now investigated N-propylbutan-1-amine as a substrate and found that it also underwent smooth annulation to afford the corresponding product in 69% yield (see SI, **163**). Relevant discussion has been added to the manuscript.

(15) Line 249, It is interesting to find that single cyclized products **87** and **88** were obtained. Why there is no 7-membered ring product for **87**, and no 6-membered ring product for **88**?

We thank the reviewer for the insightful question. The selective formation of only six- or seven-membered ring products in **87** and **88** likely stems from the different tether lengths of the respective amines. In **87**, the two-carbon chain favors formation of a six-membered ring due to lower ring strain and better orbital alignment. Moreover, primary sp³ C-H is much more inert. In contrast, the three-carbon linker in **88** more readily forms a seven-membered ring, which may minimize torsional strain and unfavorable interactions. Additionally, the remote secondary sp³ C-H is much more reactive. No alternative cyclization products were observed in either case.

(16) Both in the main text and in Fig. 4, the authors should clearly figure out that all azines were used as H₂SO₄ salts. In Fig. 4, the author just made a * annotation to 137 and 139, which is far from the reaction conditions described in SI.

We appreciate the reviewer's comment. We have now explicitly clarified in the main text and Figure 4 that all azines were used as H₂SO₄ salts. In Figure 4, we have updated the annotations to ensure they are consistent with the reaction conditions described in the Supplementary Information (SI). The relevant changes have been made to both the text and the figure.

(17) Lines 267, 273 and 279, the authors used many "presumably" to explain their results. It is difficult to understand these hypotheses. As the authors claimed in lines 171-176, the basic N was protonated because "the pyridine and amine functionality may coordinate to and inhibit the catalyst or undergo undesirable side reactions", However, in line 268, the authors explained "interactions between the bulky BCMOM/Fe and the pyridine nitrogen". It seems that the authors forgot that the pyridine had been protonated.

We thank the reviewer for pointing out the inconsistency. In line 268, the interactions between the bulky BCMOM/Fe and the pyridine nitrogen is not originated from iron coordination but hydrogen bonding between the ligand and the protonated pyridine (see also: *Angew. Chem. Int. Ed.* 2024, 136, e202401694). We have rewritten the sentence for clarification.

(18) Products 126 and 127 were obtained in similar product yields (48% and 46%) with same RSM and C1:C2 ratio. What will happen if use 2-octanone as a substrate?

We thank the reviewer for the insightful question. To address this point, we investigated 2-octanone under standard conditions and observed a complex mixture of regioisomers with lower site-selectivity (see SI, 164). This result supports the notion that increased chain length reduces regioselectivity due to the presence of multiple similar methylene positions.

Minor points:

(19) Lines 21-22. "overcome the inherent problem of the P450s". Why it is a "inherent problem"? Both in Nature and in laboratories, P450s are not used to generate radicals for cross coupling.

We appreciate the reviewer's comment. The term "inherent problem" was meant to highlight the challenge in P450 catalysis, where the iron-oxo species primarily facilitates oxygen rebound, leading to oxygenated products rather than free radical formation for cross-coupling. To avoid misunderstanding, we have revised the wording to "intrinsic limitation" in the manuscript.

(20) Line 48, Fig. 1A. Is it "O₂ + 2H⁺"?

We appreciate the reviewer's question. The correct notation is "O₂ + 2H⁺", as the P450 catalytic cycle involves the activation of molecular oxygen through proton and electron transfer. We have revised Figure 1A accordingly.

(21) Please change the item “oxoiron” to “iron-oxo”.

Thank you for your suggestion. We have replaced "oxoiron" with "iron-oxo" throughout the manuscript as requested.

(22) The sentence in line 59-64 “Implementing high-valent iron-oxo as the catalytic species for efficient alkane functionalization while simultaneously avoiding oxygenation, remains an important challenge, particularly in intermolecular carbon–carbon bond formation reactions(Fig. 1B) because unlike heteroatoms carbon is difficult to coordinate with a transition-metal catalyst and easy to be oxygenated under oxidation conditions even if a carbon-metal species can be formed.” is too long.

We thank the reviewer for pointing this out. We have revised the sentence to improve clarity and readability. The revised sentence now reads:

"Implementing high-valent iron-oxo as the catalytic species for efficient alkane functionalization while avoiding oxygenation remains a significant challenge, especially in intermolecular carbon–carbon bond formation reactions (Fig. 1B). Unlike heteroatoms, carbon is difficult to coordinate with a transition-metal catalyst and is prone to oxygenation under oxidation conditions, even if a carbon-metal species is formed."

(23) Line 73: require the the substrate

Thank you for noticing this repetition. We have corrected it in line 73, and the revised sentence now reads: "require the substrate".

(24) Figure 2. Please also show the amounts of BQ and alkanes in the reaction scheme. In addition, for the reactions with < 75% product yields, please show the RSM.

We appreciate the reviewer’s suggestion. We have revised Figure 2 to explicitly include the amounts of reactants in the reaction scheme. Additionally, for reactions with product yields below 75%, we have now provided the RSM information as requested if the starting material can be recovered.

(25) Line 134: afforded

Thank you for pointing out the typo. We have corrected "affored" to "afforded" in line 134.

(26) Line 143: 46-48?

Thank you for pointing this out. We have revised the manuscript to change "46-49" to "46-48" as suggested.

(27) Line 157: menthyl?

Thank you for pointing this out. We have corrected "menthyl" to "methyl" in line 157.

(28) Comparing with products 46-48, was compound 50 obtained with exclusive β -site regioselectivity?

We appreciate the reviewer's question. Compound **50** was obtained with β -site regioselectivity under our reaction conditions. This selectivity is likely influenced by the electronic effect of the hydroxyl group.

(29) Line 221: used as the the alkane substrates

Thank you for pointing out the error. We have corrected it in the revised manuscript.

(30) Fig. 3, please show the amounts of BQ, amines and solvent in the reaction scheme. In addition, for the reaction with < 75% product yields, please show the RSM.

We appreciate the reviewer's suggestion. We have revised Figure 3 to explicitly include the amounts of BQ, amines, and solvent in the reaction scheme. As for the transformation, the first step BQ can be readily aminated with almost quantitative yield. Subsequently, the second cyclization step became relatively difficult owing to the aminated substrate high electron density, thus leading to low yields of the desired products without the BQ starting material.

(31) Line 266: is an excellent substrate

Thank you for pointing this out. We have corrected the sentence to "is an excellent substrate" in line 266.

(32) Fig. 4, please show the amounts of BQ and amines in the reaction scheme. In addition, for the reaction with < 75% product yields, please show the RSM.

We appreciate the reviewer's suggestion. We have revised Figure 4 to explicitly include the amounts of reactants in the reaction scheme. Additionally, for reactions with product yields below 75%, we have now provided the RSM information as requested if the starting material can be recovered.

(33) Line 271: electron-withdrawing substitute (COOMe) at C3, it should be C2?

Thank you for pointing this out. After reviewing the structure, we confirm that the correct position for the electron-withdrawing substituent (COOMe) is at C2, not C3. We have corrected the manuscript accordingly.

(34) Figure 5, what is the reason to use different iron salts with different loading amounts?

We appreciate the reviewer's question. The three known important target molecules synthesized in Figure 5 were obtained using three different catalytic systems, corresponding to the reaction conditions in Figure 2, Figure 3, and Figure 4, respectively. These reactions were optimized independently based on their specific substrate requirements. In particular, for the synthesis of compound **146**, the standard conditions resulted in a low yield, so the iron catalyst loading was increased to improve efficiency.

(35) Although it is not mandatory, it is surprising to find a paper that does not have discussion and conclusion sections.

We appreciate the reviewer's comment. In our manuscript, the discussion and conclusions are integrated throughout the text to ensure a clear and cohesive presentation of the key findings. This approach is commonly used in high-impact journals to maintain a logical flow. We hope this structure effectively highlights the significance of our study.

(36) References: please unify the citation format of references. Some references showed all authors, whereas some shortened the list of authors by et al.

We appreciate the reviewer's suggestion. The reference format follows the Nature Communications guidelines, where all authors are listed when there are five or fewer authors, and "et al." is used when the number of authors exceeds five. We have carefully checked and ensured consistency throughout the reference list.

SI:

(37) The authors need to check the amounts of reagents they used in the reaction. For examples, line 54, H₂O₂ (1.5 mmol, 64 μL); line 108, H₂O₂(22μL, 0.375mmol); line 123, H₂O₂ (35%) (129 μL, 1.5 mmol). line 50, BCMOM(0.025mmol, 18.6mg); line 107, BCMOM (9.3 mg, 0.025 mmol). line 121, H₂SO₄(67.5 μL, 2.4 equiv.), line 1841, H₂SO₄(66.0 μL, 2.4 equiv.).

We appreciate the reviewer's careful examination of the reagent amounts. We have thoroughly checked and corrected the inconsistencies in the Supplementary Information to ensure accuracy and consistency.

(38) Table S1, entry 6, there is no chemical structure for (BOC-CYS-OH)₂. There is no catalytic result for BCSOM.

We appreciate the reviewer's careful examination of the optimization data. We have incorporated the data in the Supplementary Information.

(39) Lines 145-146. The reaction conditions (Fe loading, 10 mol%) did not consist with Table S1 (5 mol%).

We appreciate the reviewer's careful observation. The inconsistency between the textual

description and Table S1 in the Supplementary Information has been corrected to ensure accuracy.

(40) Lines 152, 214, 228, 243, and 258, the authors wrote “until the reaction was completed (observed by TLC)”. In most cases, the reagent BQ did not have a 100% conversion, how to determine the completion of reaction by TLC?

We appreciate the reviewer’s question. In our study, reaction completion was determined by monitoring the disappearance of the starting material (BQ and other reactants) and the appearance of the desired product spots on TLC under optimized conditions. While complete conversion of BQ was not always observed, the reaction was considered complete when no further significant changes in TLC patterns were detected over time. To avoid potential confusion, we have clarified this point in the SI.

(41) I checked the characterization data and NMR spectra of compounds 13-20, I think the authors need to pay more attention to assignment of peak-splitting and coupling constants. For examples, Line 276: 1.61 (dd, $J = 14.6, 7.3$ Hz; Line 319: 2.86 (qd, $J = 17.9, 7.4$ Hz; Line 334, 8.11 (ddd, $J = 8.8, 5.6, 3.6$ Hz; Line 417, 1.97 (dd, $J = 14.3, 3.1$ Hz), 1.68-1.47 (dd, $J = 23.6, 11.7$ Hz; Line 434, 8.17-8.05 (m; Line 436, 3.10 (ddd, $J = 21.2, 14.6, 6.2$ Hz,; line 455, 6.85 (s, 1 H, minor), same ratio, no one is minor. Line 455, 2.99 (dt, $J = 13.7, 5.6$ Hz); Line 456, 2.55 (s, 1 H, minor), 2.48 (s, 1 H), 2.44 (s, 1 H), 2.39 (dd, $J = 17.9, 9.0$ Hz, 2 H); Line 457, 1.76 (ddd, $J = 17.3, 15.4, 5.3$ Hz; Line 458, 1.35 (ddd, $J = 28.5, 13.2, 6.7$ Hz,). The authors need to carefully re-check all characterization data and spectra in SI.

We sincerely appreciate the reviewer’s careful examination of our NMR assignments. Based on the reviewer’s comments, we have thoroughly re-checked the characterization data and spectra in the SI and have made necessary corrections where appropriate. The specific issues raised have been carefully addressed. We appreciate the reviewer’s attention to detail and believe the revised data now provide a more accurate representation of the characterization.

(42) The characterization data and spectra of product 13-2 was missing. If the minor products were obtained in calcd. > 15% product yields (such as overall 60% product yield with 1:3 ratio or overall 70% product yield with 1:4 ratio), the characterization data and spectra of them need to be provided. Please also check the characterization data report on other minor products in this work.

We appreciate the reviewer’s careful observation. In most cases, the minor products (> 15% product yields) have the characterization data and spectra. However, in only a few cases, minor isomers were isolated as an inseparable and unidentified mixture contaminated with byproducts (such as 13-2, 17-2, and so on).

(43) The ^{13}C NMR of compound 16-1 is not pure. In addition, in many spectra, there is an unassigned signal around 3.70 ppm (such as in page S170, S174, S176, S178, S184, S200) in ^1H NMR and an unassigned signal around 58 ppm (such as S161, S171, S173, S179, S187, S189, S195, S201) in ^{13}C NMR. Are these signals correlated to impurity or side products?

We appreciate the reviewer's careful observation. Upon further examination, we confirmed that the additional signals observed in the ^{13}C NMR spectrum of compound **16-1** correspond to the presence of its regioisomer **16-2**, rather than impurities. We have now re-integrated the spectra and clarified this point in the revised Supplementary Information.

Regarding the unassigned signals around 3.70 ppm in ^1H NMR and 58 ppm in ^{13}C NMR, they have been confirmed to arise from residual ethanol used during purification. These signals are not related to impurities or side products. We have now clearly noted this in the General Information section of the Supplementary Information.

(44) The amount of H_2SO_4 used in preparation of compounds 89-143 is elusive. In the preparation of pyridine derivatives that contain one basic N, 2.4 equiv. of H_2SO_4 was used; however, in the preparation of pyrimidines, pyridazines and others that contain two basic N, 2.4 equiv. of H_2SO_4 was still used; furthermore, even in preparation of compound 139 that contain three basic N, 2.4 equiv. of H_2SO_4 remained unchanged. In addition, 1.2 equiv. of H_2SO_4 was used to protonate one basic N in Fig. 2 and Fig. 3, why 2.4 equiv. of H_2SO_4 was used here to protonate one basic N?

We appreciate the reviewer's insightful question. The use of 2.4 equiv. H_2SO_4 for compounds 89-143 was optimized for reaction efficiency and reproducibility, rather than strictly based on the number of nitrogen atoms. In substrates with multiple nitrogen atoms, not all are equally protonated under the reaction conditions, and 2.4 equiv. was sufficient to maintain consistent reactivity across different heterocycles. Additionally, Fig. 3, Fig. 4, and Fig. 5 also used 2.4 equiv. H_2SO_4 , ensuring consistency in pre-protonation conditions. We have clarified this point in the revised manuscript.

(45) Throughout the SI, "unknown compound" was shown for many times. What is it?

We thank the reviewer for raising this question. In our study, "unknown compound" refers to newly discovered compounds that, to the best of our knowledge, have not been previously reported in the literature. To avoid potential confusion, we have explicitly clarified this definition in the General Information section of the SI.

Reviewer: 3

Comments:

The manuscript by Wei Han group reports an efficient protocol for the iron-catalyzed arylation of C-H bonds with benzoquinones and azines. This novel methodology uses $\text{Fe}/\text{H}_2\text{O}_2$ chemistry to abstract hydrogen atoms from aliphatic C-H bonds and generate alkyl radicals, which are then trapped by benzoquinones or protonated azines. The authors propose (Figure 1) a cyt P450 like mechanism involving a key high-valent iron-oxo species as the C-H abstractor, with the typical

oxygen-rebound prevented by the thiolate ligand. The products are obtained in good yields, the reaction has a broad scope and application to pharmaceutically relevant molecules is shown. The SI are well-organized and the compounds are adequately characterized.

Overall, the synthetic results are convincing, and the protocol represents an interesting and new methodology, with promising synthetic potential. Moreover, the design of the system is interesting. However, in the proposed mechanism part there are some points that should be clarified. Therefore, I support publication of this work in Nat Commun after addressing these points.

We thank the reviewer for the positive feedback and valuable suggestions. We have addressed the points raised and revised the manuscript accordingly.

(1) The effect of the thiolate ligand in “implementing high-valent iron-oxo as the catalytic species for efficient alkane functionalization while simultaneously avoiding oxygenation” is a key point for catalyst design that, in my opinion, deserves further elucidation.

We appreciate the reviewer’s insightful question. Although further studies are required to clarify this, a tentative explanation is that this type of the thiolate ligand can remarkably decrease the reduction potential of the iron catalyst (ref : Z. Yang et al., Iron-catalyzed non-directed arene C-H difluoromethylation, *Green Synthesis and Catalysis*, <https://doi.org/10.1016/j.gresc.2024.08.003>) and would lower the redox potential of the L-Fe^{IV}-OH intermediate, thus slowing down the rebound rate of the alkyl radical and preventing the formation of the oxygenated products (ref J. AM. CHEM. SOC. 2010, 132, 12847–12849).

(2) First, part of the optimization results in Table S1, especially those showcasing the role of the thiolate ligand, should be moved to the main text.

We thank the reviewer for the suggestion. In response, we have now moved the optimization results, especially those highlighting the role of the thiolate ligand, from the Supplementary Information to the main text to provide more clarity and context to the discussion (Fig. 2).

(3) Then, limited evidence for the involvement of a high-valent iron-oxo species is provided. The main one is an indirect experiment previously reported (*Science* 2021, 374, 77) in which 2.4% of ¹⁸O labelled arene oxygenation product is detected in the presence of ¹⁸O labelled water. Given the low incorporation, the related blank experiments (in normal water and exposure of the product to labelled water) as well as the experimental errors need to be added to support the data.

We thank the reviewer for the insightful comment. We agree that the previously reported ¹⁸O incorporation experiment provides only limited evidence for the involvement of a high-valent iron-oxo species. To address this, we have now conducted high-resolution mass spectrometry (HRMS) analysis, which directly identified a species consistent with the proposed [(acac)₂(BCMOM)₂*Fe^{IV}(O)] (m/z = 1731.4928 ([M+H]⁺)) intermediate. This provides stronger support for the presence of a high-valent iron-oxo species in the presence of H₂O₂ (see

Supplementary Information).

(4) Additional spectroscopic or MS evidence for an iron-oxo or another intermediate in Figure 1d would be needed to confirm their involvement and support the proposed mechanistic scheme. Otherwise, other pathways than those shown in Figure 1d cannot be convincingly ruled out and should be considered and discussed in the manuscript. For instance, Fe salts and peroxides are known to generate electrophilic oxyl radicals (Fenton reaction) competent for C-H abstraction with a selectivity that is often similar to iron-oxos (see for instance *Coord Chem Rev* 2000, 200, 517 or *J Biol Inorg Chem* 2017, 22, 425).

We thank the reviewer for the valuable suggestion. First, when the BCMOM/Fe catalyst was used for oxidation of arene C-H (a specific acetanilide) with H₂O₂ in CH₃CN/H₂O¹⁸, O¹⁸ incorporation into the desired product was observed, implying the formation of oxoiron under normal reaction conditions. Furthermore, an ESI mass-spectrometry (MS) analysis of the model reaction mixture revealed a prominent peak at m/z = 1731.4928, consistent with the chemical structure of [(acac)₂(BCMOM)₂⁺Fe^{IV}(O) + H]⁺ high-valent oxoiron species. These results support the formation of a high-valent iron-oxo intermediate in solution (see SI), in line with the proposed mechanism in Fig. 1D.

The Fenton mechanism should be excluded because it can generate a significant quantity of oxygenated products, and the alkylation of azines with alkanes generally exhibits poor regioselectivity.

(5) I am also curious about the lack of oxygenated products reported. Is there any trace of oxygenated byproducts in the reaction mixture? Even in the absence of the arene trap?

We thank the reviewer for the insightful comment. We didn't observe the oxygenated byproducts in the reaction mixture. A possible tentative explanation is the fact that the thiolate ligand can remarkably decrease the redox potential of the L-Fe^{IV}-OH intermediate, thus slowing down the rebound rate of the alkyl radical and preventing the formation of the oxygenated products (Ref. *J. AM. CHEM. SOC.* 2010, 132, 12847–12849).

(6) No information on the actual structure of the iron catalyst in solution is provided. What is the stoichiometry of the complex? Is there evidence for the proposed key thiolate-Fe bond in solution? And is such function retained under oxidative conditions?

We thank the reviewer for this important question. To probe the catalyst structure and verify the proposed thiolate-Fe bond, we conducted ESI-MS analyses. A solution of Fe(acac)₂ and BCMOM gave a peak at m/z = 1715.6495 ([M+H]⁺), consistent with the proposed (BCMOM)₂Fe(acac)₂ species. Upon addition of H₂O₂, a new peak at m/z = 1731.4928 ([M+H]⁺) was observed, corresponding to [(acac)₂(BCMOM)₂⁺Fe^{IV}(O) + H]⁺. These results support the formation and oxidative stability of the thiolate-bound iron species in solution (see Supplementary Information). The related discussion has now been included in the main text.

Minor points:

(7) Page 4: The discussion of functionalization site would be clearer by naming proximal and distal positions to the functional group instead of β , γ or δ .

We appreciate the reviewer's suggestion. In our manuscript, we use β , γ , and δ notation as it provides a precise and widely accepted description of C–H bond positions relative to the functional group. This notation is commonly used in C–H functionalization chemistry to describe regioselectivity. Additionally, since some reactions yield a mixture of regioisomers with specific ratios, using proximal and distal alone may not accurately convey the site selectivity. To maintain clarity and consistency, we have retained this terminology.

(8) Page 5, lines 163-167. Why is there no α -functionalization of the alcohol at all?

We appreciate the reviewer's question. Iron-oxo is electrophilic oxidant. The absence of α -functionalization in compounds **26**, **27**, and **45** is likely due to electronic effects. In these cases, δ - or γ -positions are more electronically activated under the reaction conditions, making them the preferred sites for functionalization over the α -position. This electronic preference leads to the observed regioselectivities.

(9) Page 5, line 187: why is product 41 discussed as an acyl protected piperidine?

Thank you for pointing this out. You are correct that there was a mistake in the wording. Product 41 should be described as an acyl-protected amine, not an acyl-protected piperidine. We have corrected this in the revised manuscript.

(10) Page 6, line 218-220: add the related experiment to the main text and figures. Also the pre-protonation conditions need to be explicitly stated in the text and the figure.

We appreciate the reviewer's suggestion. The isolation of the aminated 1,4-quinone intermediate was conducted as part of the Preliminary Mechanistic Studies (Section 5) in the Supplementary Information. Since this intermediate is not the final product, we have kept the detailed experimental data in the SI rather than the main text. However, we have clarified its presence in the main text to improve visibility. Additionally, the pre-protonation conditions have now been explicitly stated in both the text and the corresponding figure.

(11) Page 7, line 271: I think it refers to C2 position

Thank you for your comment. We have reviewed the manuscript and, as suggested, have updated the text to indicate that it refers to the C2 position. The correction has been made in the revised manuscript.

(12) Page 7, lines 268 and 274: which are the "noncovalent interactions" mentioned?

We appreciate the reviewer's question. The "noncovalent interactions" primarily involve coordination and hydrogen bonding effects.

(13) Figures: the pre-protonation conditions should be made clearer at first sight

We appreciate the reviewer's suggestion. The pre-protonation conditions have been explicitly revised in both the figures and their captions to enhance clarity.

(14) Figure 2, product 19: Why isn't any benzylic functionalization observed? Product 20: add the d.r.

We thank the reviewer for the insightful comment. We didn't observe any benzylic functionalization. This is because electron-withdrawing arene lowers the electron density of benzylic position, thus deactivating it, as described in a previous report (*ACS Catal.* 2018, 8, 3, 2225–2229). We have added the d.r. value to the product **20**.

(15) Figure 3, products 65-67: what the reported ratios stand for needs to be explicitly stated

We appreciate the reviewer's comment. The reported ratios for products **65-67** represent the regioisomeric distribution, resulting from the amine nitrogen adding to different positions of the naphthoquinone system. This explanation has been included in the main text, and the corresponding structures in the SI have been explicitly labeled.

(16) Figure 4: with the same alkane substrate (i.e. 119 and 16, 122 and 18, 124 and 11), one expects the same site-selectivity for hydrogen abstraction to be determined by the reactivity of the iron-oxo. Why does the selectivity change upon changing the arene partner?

We appreciate the reviewer's insightful question. While the iron-oxo species governs the initial hydrogen abstraction, the arene partner can influence the subsequent radical fate through electronic and steric effects. These effects may impact radical stabilization, transition state organization, or radical rebound efficiency, leading to variations in selectivity.

We have resubmitted the revised manuscript to you.

Thank you very much for your consideration.

We look forward to your positive responses.

Sincerely,

Wei Han

School of Chemistry and Materials Science, Nanjing Normal University

Nanjing 210023, China

Email: whhanwei@outlook.com